# The Hidden Life of Tokens: Reducing Hallucination of Large Vision-Language Models via Visual Information Steering

Zhuowei Li [1]   Haizhou Shi [1]   Yunhe Gao [1 2]   Di Liu [1]   Zhenting Wang [1]   Yuxiao Chen [1]   Ting Liu [3]   Long Zhao [3]
Hao Wang [1]   Dimitris N. Metaxas [1]

## Abstract

Large Vision-Language Models (LVLMs) can reason effectively over both textual and visual inputs, but they tend to hallucinate syntactically coherent yet visually ungrounded contents. In this paper, we investigate the internal dynamics of hallucination by examining the tokens logits ranking throughout the generation process, revealing three key patterns in how LVLMs process information: *(1) gradual visual information loss* – visually grounded tokens gradually become less favored throughout generation, and *(2) early excitation* – semantically meaningful tokens achieve peak activation in the layers earlier than the final layer. *(3) hidden genuine information* – visually grounded tokens though not being eventually decoded still retain relatively high rankings at inference. Based on these insights, we propose **VISTA** (**V**isual **I**nformation **S**teering with **T**oken-logit **A**ugmentation), a training-free inference-time intervention framework that reduces hallucination while promoting genuine information. VISTA works by combining two complementary approaches: reinforcing visual information in activation space and leveraging early layer activations to promote semantically meaningful decoding. Compared to existing methods, VISTA requires no external supervision and is applicable to various decoding strategies. Extensive experiments show that VISTA on average reduces hallucination by about 40% on evaluated open-ended generation task, and it consistently outperforms existing methods on four benchmarks across four architectures under three decoding strategies. Code is available at https://github.com/LzVv123456/VISTA.

*Equal contribution  [1]Rutgers University [2]Stanford University  [3]Google DeepMind. Correspondence to: Zhuowei Li <zhuowei.li@rutgers.edu>.

*Proceedings of the 42nd International Conference on Machine Learning*, Vancouver, Canada. PMLR 267, 2025. Copyright 2025 by the author(s).

## 1. Introduction

Large Vision-Language Models (LVLMs) (Dai et al., 2023; Bai et al., 2023; Chen et al., 2023; Zhu et al., 2023; Liu et al., 2024c) have revolutionized multimodal AI by enabling seamless integration of visual and textual information, powering applications from interactive assistance to autonomous systems (Lin et al., 2023; Yang et al., 2024; Lai et al., 2024). However, LVLMs frequently hallucinate semantically coherent yet visually ungrounded contents, hindering their reliability in real-world applications.

Though LVLM hallucination is considered multifaceted (Liu et al., 2024d), a critical cause stems from the overwhelming influence of language priors over visual contexts, and has been studied from the perspective of attention patterns (Huang et al., 2024; Liu et al., 2024g) and distribution divergence within logits space (Leng et al., 2024; Favero et al., 2024). Despite these insights, it remains unclear how hallucination emerges and propagates internally during the course of generation.

**Inspecting Token Dynamics in LVLMs.** In this work, we take a novel perspective by examining LVLM's generation dynamics through the lens of token logits ranking. Given an image and its corresponding description (produced by an LVLM), we identify three categories of tokens (elaborated in Sec. 3.4):

- *Hidden Genuine Tokens* – tokens that are missing in generated contents yet clearly visible from visual input;
- *Decoded Genuine Tokens* – tokens that appear in continuation with visual groundings;
- *Hallucinated Tokens* – tokens extracted from the hallucinated contents within generation.

We then track each token type's corresponding logits rankings throughout the generation across temporal (Fig. 1 left) and layer sequences (Fig. 1 right). Our analysis makes three prominent observations:

- **(OBS-1) Gradual Visual Information Loss.** As generation progresses, genuine token rankings gradually decline while hallucinated tokens are surfaced (see Fig. 1 left and Fig. 2). This aligns with recent findings (Yue

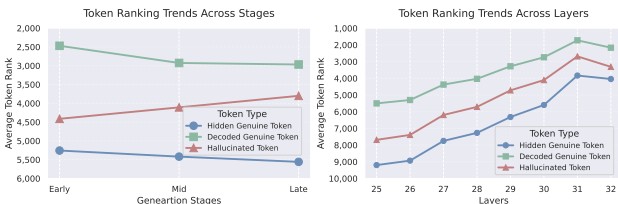

*Figure 1.* Analysis of token logits ranking patterns across 500 randomly selected images from MSCOCO dataset. Higher ranking indicates higher generation probability. **Left:** Average token ranking from the last five layers, showing temporal progression across early, mid, and late generation stages. **Right:** Layer-wise evolution of token rankings averaged across all time steps, demonstrating early-excitation phenomenon.

et al., 2024; Favero et al., 2024), exhibiting increasing hallucination in late generation phase. We hypothesize that this occurs as accumulated language priors in residual streams progressively dilute visual information, leading to syntactically coherent but visually ungrounded generation.

- **(OBS-2) Early Excitation of Semantically Meaningful Tokens.** Semantically meaningful tokens [1] exhibit peak excitation in penultimate layer (Fig. 1 right) or within a window of layers preceding the final layer (Fig. 2). In contrast, the final layer prioritizes functional tokens like "this", "a", and other stopwords, suggesting that the model's decision process may overemphasize syntactic elements in its final stage.

- **(OBS-3) Hidden Genuine Information.** LVLMs may perceive more visual clues than they express. We observe that hidden genuine tokens, though not eventually decoded, consistently maintain relatively high rankings (around 5K in a 32K vocabulary) during the course of generation (see Fig. 1).

**Reducing Hallucination of LVLMs.** Inspired by above findings, we propose **VISTA** (**V**isual **I**nformation **S**teering with **T**oken-logit **A**ugmentation), a simple and novel training-free framework that can be applied on top of various decoding methods to reduce hallucination while promoting genuine information. VISTA introduces two complementary modules: *Visual Steering Vector (VSV)* that counteracts gradual visual information loss by extracting and reinforcing visual cues in activation space, and *Self-Logits Augmentation (SLA)* which utilizes early excitation patterns to prioritize semantically meaningful tokens. VSV and SLA work synergistically and can effectively mitigate hallucination in LVLMs.

**Technical Contributions.** This study presents the first systematic investigation of token dynamics in LVLMs through the lens of token logits ranking, revealing novel insights into how visual information is processed and potentially

---

[1]We refer to categorized tokens as semantically meaningful tokens since we primarily focus on identifying objects, attributes, and relations as detailed in Appendix A.

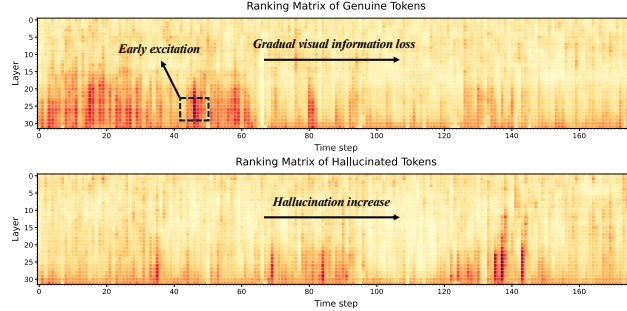

*Figure 2.* Token ranking heatmaps for a representative image, demonstrating the evolution of token rankings across model layers (vertical axis) and generation steps (horizontal axis). Darker colors indicate higher ranking. The visualization reveals both gradual visual information loss and early excitation phenomena.

lost during generation. Building on these insights, we propose VISTA, an inference-time intervention framework that incorporates two complementary modules. The effectiveness of VISTA is validated through comprehensive experiments across multiple architectures (e.g., LLaVA, Shikra, MiniGPT-4, InstructBLIP) and evaluation protocols (open-ended generation, visual question answering), demonstrating significant reduction in hallucination (up to around 40% for open-ended generation). Our approach is notably efficient as it requires no additional training or model modifications, making it readily applicable to existing LVLM deployments.

## 2. Methodology

In this section, we first establish the notions and knowledge foundation in Sec. 2.1, followed by the elaboration of token ranking analysis in Sec. 2.2. We then present the proposed VSV and SLA methods in Sec. 2.3 and Sec. 2.4, respectively.

### 2.1. Preliminaries

**Conditional Generation of LVLMs.** Suppose that an LVLM consists of a vision encoder and a cross-modal interface that projects visual inputs into a sequence of visual tokens $\boldsymbol{X}_v$. Given an input image, the complete prompt tokens $\boldsymbol{X}_c$ are constructed by concatenating system message tokens $\boldsymbol{X}_s$ (can be empty), visual tokens $\boldsymbol{X}_v$, and query tokens $\boldsymbol{X}_q$: $\boldsymbol{X}_c = \texttt{concat}(\boldsymbol{X}_s, \boldsymbol{X}_v, \boldsymbol{X}_q)$. At each time step $t$, the model samples a new token $x_t$ according to the probability distribution conditioned on both the input context $\boldsymbol{X}_c$ and previously generated tokens $\boldsymbol{X}_{<t} = \{x_i\}_{i=1}^{t-1}$:

$$x_t \sim p(\boldsymbol{x}_t | \boldsymbol{X}_c, \boldsymbol{X}_{<t}) = \text{softmax}(\mathcal{H}(\boldsymbol{h}_{t-1}^L)), \quad (1)$$

where $\mathcal{H}$ denotes the model's head layer and $\boldsymbol{h}_{t-1}^L$ indicates the hidden state from the last layer $L$ at time step $t-1$. Above formulation suggests that the dilution or insufficiency of visual information in $\boldsymbol{h}_{t-1}^L$ can bias the generation towards hallucination.

**Residual Stream.** Taking the mathematical interpretation from Elhage et al. (2021), we view layer-wise hidden states as residual streams that evolve recursively:

$$\boldsymbol{h}_t^l = \boldsymbol{h}_t^{l-1} + \boldsymbol{a}_t^l + \boldsymbol{m}_t^l, \tag{2}$$

where $l$ is the layer index, $\boldsymbol{a}_t^l$ and $\boldsymbol{m}_t^l$ represent the output activation of integrated multi-head attention (MHA) layer and feed-forward network (FFN), respectively. Within this framework, MHAs facilitate information fusion across different residual streams, while FFNs access and integrate learned parametric knowledge (Geva et al., 2021; Dai et al., 2022). Residual stream provides a natural interface for monitoring and controlling information flow, making it particularly suitable for hallucination analysis and mitigation.

**Logit Lens.** The head layer $\mathcal{H}$ is by default applied on top of last layer hidden states $\boldsymbol{h}^L$. However, thanks to the gradual evolvement of hidden states within residual streams (Chuang et al., 2024), applying $\mathcal{H}$ to hidden states of earlier layers $l < L$ remains effective, even without additional training (Gurney, 2023). This practice is commonly referred to as "logit lens" and can be used to decipher intermediate states.

## 2.2. Token Ranking Analysis

To systematically investigate how visual information is processed during generation, we propose a token ranking analysis framework that tracks the relative importance of different tokens throughout the generation process.

**Identification of Target Tokens.** For each given image-description pair where the text description is generated by an LVLM (e.g., LLAVA-1.5 (Liu et al., 2024a)), we utilize gpt-4o (Hurst et al., 2024) as the oracle model to identify three categories of words referencing both visual and textual contents. A word is a

- *decoded genuine word* if it appears in the continuation and align with visual evidence;
- *hidden genuine word* if it is visually evident but not included in the continuation;
- *hallucinated word* if it appears in continuation but lacks visual grounding.

Collected words are then tokenized to form our analysis sets. Implementation details are included in Appendix A.

**Token Ranking via Logit Lens.** To analyze token dynamics during generation, we apply the logit lens $\mathcal{H}(\boldsymbol{h}_t^l)$ to each layer $l$ and time step $t$. Given token $x$, we calculate its ranking position among all possible tokens according to:

$$R_t^l(x) = \text{rank}(\mathcal{H}(\boldsymbol{h}_t^l), x), \tag{3}$$

where $R_t^l(x)$ represents the position of token $x$ in the probability-ordered sequence of all tokens at time step $t$

and layer $l$. A lower rank indicates higher probability. This operation produces a 2D ranking matrix for each token with vertical and horizontal axes indicating layers and time steps, respectively. We aggregate these matrices across tokens within each category to obtain category-specific ranking patterns, as shown in Fig. 2.

For temporal analysis, we quantize time sequence into three equal-sized buckets, i.e., early, mid, and late, and compute average rankings within each bucket across 500 randomly sampled images from MS COCO dataset (Lin et al., 2014). This temporal view (Fig. 1 left) reveals that visually grounded information is gradually sinking while hallucinated contents are surfaced. We further average rankings across all time steps within a layer to provide a layer-wise perspective (Fig. 1 right), which exhibits that semantically meaningful tokens achieve peak excitation in the penultimate layer. We refer readers to Appendix A.3 for a discussion of why token ranking analysis is desirable and its limitations.

## 2.3. Visual Steering Vector (VSV)

Being aware of the challenge from gradual visual information loss, it is of critical importance to retain visual cues throughout the generation. A promising method involves increasing attention weights distributed on visual tokens (Liu et al., 2024g). Nevertheless, this operation simultaneously introduces undesired parametric priors cumulated in residual streams of visual tokens. Drawing inspiration from steering vectors in LLMs (Turner et al., 2023; Zou et al., 2023; Liu et al., 2024e; Li et al., 2024), we propose Visual Steering Vector (VSV) to steer the generation of LVLM towards the direction with visual groundings without amplifying inherent language biases.

**VSV Construction.** The core logic behind VSV is to extract a directional vector within activation space without introducing disturbing language priors. To this end, we construct VSV via a contrastive process using paired context sequences: a "positive" context $\boldsymbol{X}_p = \texttt{concat}(\boldsymbol{X}_s, \boldsymbol{X}_v, \boldsymbol{X}_q)$ containing visual tokens $\boldsymbol{X}_v$, and a "negative" counterpart $\boldsymbol{X}_n = \texttt{concat}(\boldsymbol{X}_s, \boldsymbol{X}_q)$ that discards visual tokens while preserving other elements. Both sequences are processed by a vectorization function $\mathcal{F}$, which forwards the given token sequence through the LVLM and takes the residual stream from the last token. Visual steering vector (VSV) can be computed as:

$$\boldsymbol{V}_{\text{steer}} = \boldsymbol{V}_p - \boldsymbol{V}_n = \{\boldsymbol{v}_{\text{steer}}^l\}_{l=1}^L, \tag{4}$$

where $\boldsymbol{V}_p = \mathcal{F}(\boldsymbol{X}_p)$ and $\boldsymbol{V}_n = \mathcal{F}(\boldsymbol{X}_n)$. Here, $\boldsymbol{v}_{\text{steer}}^l$ refers to the steering vector for layer $l$.

**Inference-time Intervention.** During inference, we inject the visual steering vector into the residual stream at each

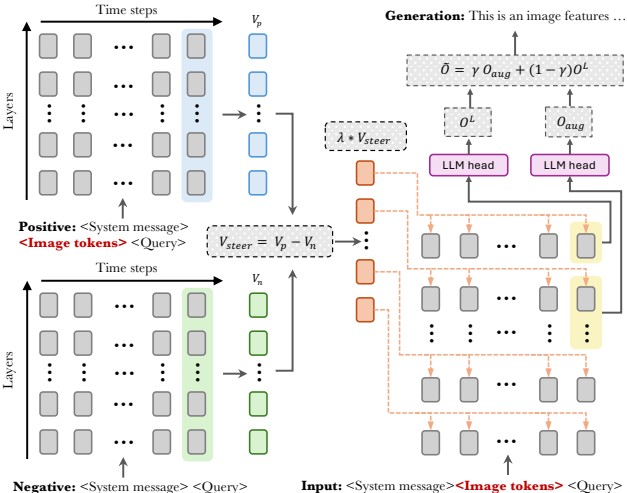

*Figure 3.* Architectural overview of VISTA. VISTA introduces two complementary mechanisms: VSV extracts and reinforces visual grounding information ($V_s$) at inference, and SLA leverages early-layer semantic information to guide token generation. *Note: While three separate forward passes are shown for illustration purpose, they can be avoided in implementation.*

generation step:

$$\tilde{\boldsymbol{h}}_t^l = \boldsymbol{h}_t^l + \lambda \boldsymbol{v}_s^l, \quad l \in [1, L], \tag{5}$$

with $\lambda$ controlling the intervention strength, balancing visual fidelity against natural generation. To maintain stability, we normalize the modified hidden states:

$$\tilde{\boldsymbol{h}}_t^l = \tilde{\boldsymbol{h}}_t^l \cdot \frac{\|\boldsymbol{h}_t^l\|_2}{\|\tilde{\boldsymbol{h}}_t^l\|_2}, \quad l \in [1, L]. \tag{6}$$

Unlike steering vectors in LLMs that capture abstract concept in an amortized fashion leveraging a group of contrastive pairs (Zou et al., 2023), VSV computes the steering vector per image basis to preserve crucial visual details unique to each input image.

**Remark on Difference with Existing Contrastive Strategy.** Existing contrastive strategy (Li et al., 2022; Leng et al., 2024) necessitates a second negative logits distribution to contrast the final layer logits at all generation steps. VSV, on the other hand, extracts a steering vector priori using only the context tokens, and is applied to all layers (Fig. 3). In practice, formatting prompt and general negative prompt (e.g., `Describe the image in detail.`) can be forwarded only once and cached for future usage, making VSV remarkably efficient (see Table 5).

### 2.4. Self-Logits Augmentation (SLA)

Motivated by early excitation phenomenon (Fig. 1 right), where semantically meaningful tokens show stronger activation in penultimate layer, we propose Self-Logits Augmentation (SLA) to promote the decoding of such tokens.

**Augmentation Logits.** To elicit the rich semantic information present in the late layers, we calculate "augmentation logits" $\boldsymbol{o}_t^{\text{aug}}$ at each decoding time step $t$. These are obtained by applying the head layer $\mathcal{H}$ to the hidden states of the $w$ layers prior to the final layer and averaging their logits:

$$\boldsymbol{o}_t^{\text{aug}} = \frac{1}{w} \sum_{l=L-w}^{L-1} \mathcal{H}(\boldsymbol{h}_t^l), \tag{7}$$

where $w$ indicates the window size. In practice, larger $w$ leads to improved performance (see Table 4). We hypothesize that this is due to the ever-evolving logits distributions within late layers, especially for those semantically meaningful tokens (Chuang et al., 2024), and applying a larger $w$ smooths the distribution and provides nuanced information.

**Logits Ensemble.** Augmentation logits are then integrated with the final layer logits through weighted aggregation:

$$\tilde{\boldsymbol{o}}_t = (1 - \gamma) \cdot \boldsymbol{o}_t^L + \gamma \cdot \boldsymbol{o}_t^{\text{aug}}, \tag{8}$$

where $\boldsymbol{o}_t^L = \mathcal{H}(\boldsymbol{h}_t^L)$ is the logits of the last layer, and $\gamma \in [0, 1]$ is a constant coefficient controlling the influence of early-layer information–when $\gamma = 0$, the model reduces to standard generation, while $\gamma = 1$ would rely entirely on early-layer logits. The next token is then sampled from the output distribution $x_t \sim \text{softmax}(\tilde{\boldsymbol{o}}_{t-1})$. Notably, SLA seeks to promote the decoding of more semantically meaningful tokens than truthful tokens as done in DoLa (Chuang et al., 2024). It works synergistically with the VSV module, enhancing overall performance (see Fig. 6).

## 3. Experiments

In this section, we empirically validate VISTA across four architectures, three decoding strategies, and four benchmarks. We first present the experimental configuration (Sec. 3.1), followed by an extensive evaluation on hallucination-specific and general-purpose benchmarks (Sec. 3.2 and 3.3). We then analyze VISTA's effectiveness in addressing the observed phenomena (Sec. 3.4) and conclude with comprehensive ablation studies (Sec. 3.5).

### 3.1. Experimental Setup

**Model Architectures.** We evaluate VISTA on four representative LVLMs with distinct architectural designs: **LLAVA-1.5** (Liu et al., 2024a) and **Shikra** (Chen et al., 2023), which employ linear projections for visual-textual alignment, and **MiniGPT-4** (Zhu et al., 2023) and **InstructBLIP** (Dai et al., 2023), which utilize Q-former (Li et al., 2023a) for cross-modal interaction.

**Decoding Strategies.** To demonstrate VISTA's versatility as an inference-time intervention method, we verify it across three widely used decoding protocols: (1) **greedy decoding**,

*Table 1.* CHAIR hallucination evaluation results. We compare VISTA to state-of-the-art training-free methods that do not rely on external supervision. Maximum new token is set to 512. **Best** and second best results are bolded and underlined, respectively. "-" indicates the result is not supported by released implementation.

| Decoding | Method | LLAVA-1.5 (Liu et al., 2024a) | | MiniGPT-4 (Zhu et al., 2023) | | Shikra (Chen et al., 2023) | | InstructBLIP (Dai et al., 2023) | |
|---|---|---|---|---|---|---|---|---|---|
| | | CHAIR$_S$ ↓ | CHAIR$_I$ ↓ | CHAIR$_S$ ↓ | CHAIR$_I$ ↓ | CHAIR$_S$ ↓ | CHAIR$_I$ ↓ | CHAIR$_S$ ↓ | CHAIR$_I$ ↓ |
| Greedy | Vanilla | 46.4 | 12.1 | 35.2 | 10.7 | 56.8 | 14.8 | 38.0 | 10.7 |
| | DoLa (Chuang et al., 2024) | 45.4 | 11.9 | - | - | 60.0 | 15.1 | - | - |
| | VCD (Leng et al., 2024) | 47.4 | 13.0 | - | - | - | - | 45.8 | 12.8 |
| | PAI (Liu et al., 2024g) | 22.8 | 7.0 | 29.2 | 10.9 | 40.8 | 11.0 | - | - |
| | **VISTA** (ours) | **20.4** | **6.9** | **19.8** | **6.0** | **31.4** | **9.7** | **27.4** | **8.1** |
| Beam Search | Vanilla | 49.0 | 12.5 | 33.0 | 11.0 | 53.8 | 14.4 | 37.8 | 10.7 |
| | VCD (Leng et al., 2024) | 49.8 | 12.4 | - | - | - | - | 49.2 | 13.7 |
| | OPERA (Huang et al., 2024) | 45.2 | 12.4 | 26.8 | 9.3 | 39.6 | 12.2 | 50.2 | 13.9 |
| | PAI (Liu et al., 2024g) | 22.3 | 6.8 | 31.6 | 11.2 | 41.6 | 10.4 | - | - |
| | **VISTA** (ours) | **17.4** | **6.3** | **18.4** | **6.4** | **32.2** | **9.5** | **26.8** | **7.8** |
| Nucleus Sampling | Vanilla | 53.2 | 15.1 | 34.8 | 11.2 | 56.4 | 15.9 | 46.6 | 13.1 |
| | DoLa (Chuang et al., 2024) | 47.2 | 14.0 | - | - | 56.6 | 16.3 | - | - |
| | VCD (Leng et al., 2024) | 60.8 | 16.2 | - | - | - | - | 57.0 | 16.0 |
| | PAI (Liu et al., 2024g) | 30.2 | 10.3 | 31.8 | 13.2 | 43.2 | 12.0 | - | - |
| | **VISTA** (ours) | **24.0** | **8.2** | **18.4** | **6.4** | **31.8** | **9.7** | **29.4** | **9.1** |

which selects the highest probability token at each step, (2) **beam search** with a beam size of 5, maintaining multiple generation hypotheses, and (3) **nucleus sampling** with top-p=0.9. Temperature is fixed at 1.0 for all scenarios.

**Baselines.** Besides three vanilla decoding strategies, we compare VISTA with several SoTA hallucination mitigation methods that can operate without external supervision. **DoLa** (Chuang et al., 2024) signifies a internal (across layers) contrastive strategy within logits space, while **VCD** (Leng et al., 2024) presents a parallel contrastive method across time steps. **OPERA** (Huang et al., 2024) is a powerful technique tailored for the beam search. **PAI** (Liu et al., 2024g) is another inference-time intervention method that is most comparable to ours. We reproduce all baseline results using identical evaluation data and settings (e.g., prompt, temperature). *Methods without official supports for certain architectures and decoding strategies are omitted to prevent implementation bias.*

**Implementation Details.** We employ the following configuration across all experiments unless stated otherwise. The VSV strength parameter $\lambda$ is set to 0.17 for both LLAVA-1.5 and InstructBLIP, 0.1 for MiniGPT-4, and 0.12 for Shikra. The SLA mixing coefficient $\gamma$ is consistently set to 0.3, with a window size $w = 5$ for aggregating early-layer logits. We search the hyperparameters of $\lambda$ and $\gamma$ on a holdout validation set containing 100 images from MSCOCO to balance between generation quality and hallucination reduction.

### 3.2. Results on Object Hallucination Benchmarks

We first evaluate VISTA on two widely adopted benchmarks that assess object hallucination: CHAIR (Rohrbach et al., 2018) for open-ended generation and POPE (Rohrbach et al., 2018) for targeted visual question answering.

**CHAIR Evaluation.** Caption Hallucination Assessment with Image Relevance (CHAIR) (Rohrbach et al., 2018) provides a systematic framework for evaluating object hallucination in image captioning tasks. CHAIR assesses caption accuracy by comparing mentioned objects against ground-truth labels, with hallucinations defined as objects present in captions but absent from ground truth. The metric operates at two levels: instance-level (CHAIR$_I$) and sentence-level (CHAIR$_S$): $\text{CHAIR}_I = \frac{|\{\text{hallucinated object}\}|}{|\{\text{object}\}|}$ and $\text{CHAIR}_S = \frac{|\{\text{caption w/ hallucinated objects}\}|}{|\{\text{caption}\}|}$. Following established protocol (Huang et al., 2024; Liu et al., 2024g), we evaluate on 500 randomly sampled images from MSCOCO 2014 validation set, using the prompt "`Please help me describe the image in detail`" with maximum generation length of 512 tokens.

Results in Table 1 show that VISTA significantly reduces hallucination in open-ended generation task, outperforming existing inference-time intervention method and other contrastive decoding methods by a substantial margin. VISTA brings around 40% relative improvement upon corresponding vanilla decoding methods. Notably, while PAI (Liu et al., 2024g) shows less efficacy for sampling-based decoding, VISTA excels across all decoding strategies. We attribute this robust performance to the contrastive design and the choice of activation space steering which is not hinged with any specific decoding strategy.

**POPE Evaluation.** The Polling-based Object Probing Evaluation (POPE) (Rohrbach et al., 2018) examines object hallucination through targeted visual questions of the form "`Is there a <object> in the image?`". The benchmark comprises three splits of increasing difficulty: random objects selected from a general vocabulary, frequently occurring objects chosen from common categories, and adversarially selected objects that are contextually plausible but absent from images. We evaluate on the COCO subset, reporting average accuracy and F1 scores across all splits. Since POPE evaluation is formulated as short VQA format and the response is simply `Yes` or `No`, the gradual visual

*Table 2.* Evaluation results on POPE benchmark across four LVLMs. Results show averaged accuracy and F1 scores computed across random, popular, and adversarial object splits. **Best** and second best results are bolded and underlined, respectively.

| Decoding | Method | LLAVA-1.5 (Liu et al., 2024a) | | MiniGPT-4 (Zhu et al., 2023) | | Shikra (Chen et al., 2023) | | InstructBLIP (Dai et al., 2023) | |
|---|---|---|---|---|---|---|---|---|---|
| | | Avg. Accuracy ↑ | Avg. F1 ↑ | Avg. Accuracy ↑ | Avg. F1 ↑ | Avg. Accuracy ↑ | Avg. F1 ↑ | Avg. Accuracy ↑ | Avg. F1 ↑ |
| Greedy | Vanilla | 84.79 | 85.61 | 76.76 | 76.82 | 81.32 | 82.01 | 84.36 | 84.64 |
| | DoLa (Chuang et al., 2024) | 84.92 | 85.67 | - | - | 81.13 | 81.94 | - | - |
| | VCD (Leng et al., 2024) | 84.80 | 85.65 | - | - | - | - | 84.81 | 85.28 |
| | PAI (Liu et al., 2024g) | 85.85 | 86.08 | 75.64 | 77.57 | 81.30 | 80.81 | - | - |
| | **VISTA** (ours) | **86.15** | **86.29** | **77.06** | **77.80** | **82.44** | **82.47** | **84.87** | **84.95** |
| Beam Search | Vanilla | 85.45 | 84.93 | 73.68 | 72.40 | 81.73 | 82.10 | 84.38 | 83.71 |
| | VCD (Leng et al., 2024) | 85.85 | 85.90 | - | - | - | - | 84.90 | 84.43 |
| | OPERA (Huang et al., 2024) | 85.68 | 85.83 | 74.81 | 75.42 | 82.18 | 82.49 | 85.31 | 85.51 |
| | PAI (Liu et al., 2024g) | **86.27** | 85.91 | 73.83 | 74.63 | 81.90 | 81.08 | - | - |
| | **VISTA** (ours) | 85.83 | **85.95** | **75.96** | **77.17** | **82.54** | **82.52** | **85.78** | **85.74** |
| Nucleus Sampling | Vanilla | 81.26 | 82.40 | 60.56 | 62.04 | 78.94 | 80.18 | 78.83 | 79.74 |
| | DoLa (Chuang et al., 2024) | 81.20 | 82.44 | - | - | 79.49 | 80.72 | - | - |
| | VCD (Leng et al., 2024) | 81.08 | 82.22 | - | - | - | - | 79.61 | 80.43 |
| | PAI (Liu et al., 2024g) | 81.92 | 83.16 | 61.26 | 63.40 | 79.25 | 79.87 | - | - |
| | **VISTA** (ours) | **85.35** | **85.54** | **66.96** | **68.05** | **81.01** | **81.15** | **83.11** | **83.27** |

*Table 3.* Overall performance scores on MME full evaluation set. Higher scores indicate better general capability across perception, reasoning, and knowledge-based tasks.

| Decoding | Method | LLAVA-1.5 | MiniGPT-4 | Shikra | InstructBLIP |
|---|---|---|---|---|---|
| Greedy | Vanilla | 1752.35 | 969.93 | 1101.50 | 1355.25 |
| | **VISTA** | **1771.87** | **1041.66** | **1256.22** | **1364.05** |
| Beam | Vanilla | 1749.57 | 869.74 | 1223.44 | 1357.02 |
| | **VISTA** | **1763.15** | **1062.48** | **1323.25** | **1366.57** |
| Nucleus | Vanilla | 1625.22 | 845.30 | 1069.60 | 1397.71 |
| | **VISTA** | **1738.56** | **1069.37** | **1254.31** | **1447.36** |

information loss is not evident. We therefore adjust VSV strength to $\lambda = 0.01$ to reduce VSV's impact.

Results in Table 2 demonstrate VISTA's consistent superiority across models and decoding strategies. Under greedy decoding, VISTA achieves average accuracies of 86.15%, 77.06%, 82.44%, and 84.87% for LLAVA-1.5, MiniGPT-4, Shikra, and InstructBLIP respectively, consistently outperforming both vanilla decoding and PAI. The improvements are particularly pronounced in nucleus sampling, where VISTA achieves significant gains over vanilla decoding across all models: from 81.26% to 85.35% (+4.09%) for LLAVA-1.5, 60.56% to 66.96% (+6.40%) for MiniGPT-4, and similarly substantial improvements for other architectures. These results highlight VISTA's particular efficacy in excavating genuine information under stochastic sampling settings, while maintaining strong performance in deterministic decoding strategies like greedy and beam search.

### 3.3. Results on Comprehensive Benchmarks

We further validate VISTA on MMHal-Bench (Sun et al., 2023) and MME (Fu et al., 2023), two challenging benchmarks that examine diverse aspects of model behavior.

**MMHal-Bench Evaluation.** MMHal-Bench (Sun et al., 2023) provides a specialized framework for assessing hallucination in LVLMs through 96 carefully designed image-question pairs. The benchmark spans eight distinct categories: object attributes (ATTR), adversarial objects (ADV), comparisons (COMP), counting (COUNT), spatial relations

(SPAT), environmental inferences (ENV), holistic descriptions (HOL), and others (OTHER). Unlike conventional VQA evaluations, MMHal-Bench emphasizes logical reasoning and complex visual understanding, providing a rigorous test of hallucination mitigation in challenging scenarios. Model responses are evaluated using GPT-4 for alignment with ground-truth answers. We compare VISTA with PAI and vanilla decoding methods for this evaluation.

The results in Fig. 4 demonstrate VISTA's consistent effectiveness across all evaluated LVLMs. Compared to vanilla methods, VISTA achieves substantial improvements on average scores, with LLAVA-1.5 and InstructBLIP showing the most pronounced gains (∼20% and ∼30% relative improvement, respectively). VISTA particularly excels in challenging categories such as environmental inference (ENV), attribute perception (ATTR) and counting (COUNT), where visual grounding is crucial for accurate responses. While PAI shows competitive performance on specific architectures, VISTA maintains more consistent improvements across both model architectures and question types. This robust performance across diverse tasks indicates that VISTA effectively addresses hallucination while preserving general visual-language capabilities. Results under beam search and nucleus sampling (see Appendix B) exhibit similar performance patterns, confirming VISTA's robustness across different decoding strategies.

**MME Evaluation.** To assess whether hallucination mitigation affects general model capabilities, we evaluate VISTA on MME (Fu et al., 2023), a comprehensive benchmark encompassing 14 distinct visual-language abilities. MME tests perception, reasoning, and knowledge integration through carefully curated image-question pairs, providing a holistic view of model performance. We conduct full-set evaluation across all architectures and decoding strategies.

As shown in Table 3, VISTA, although initially crafted for addressing hallucination, demonstrates broad benefits beyond its primary objective. Performance improvements are

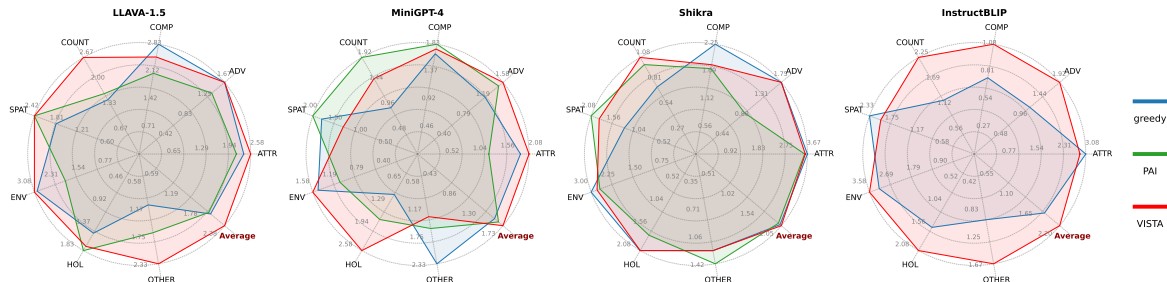

*Figure 4.* Performance comparison on MMHal-Bench across different question categories: attributes (ATTR), adversarial objects (ADV), comparisons (COMP), counting (COUNT), spatial relations (SPAT), environmental inference (ENV), holistic descriptions (HOL), and others (OTHER). Scores are computed using GPT-4 evaluation protocol.

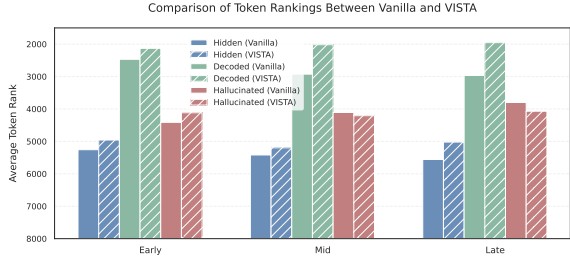

*Figure 5.* Cross-stage token ranking comparison between greedy and VISTA on LLAVA-1.5. VISTA effectively promotes the ranking of genuine tokens while depressing hallucination tokens.

consistent across architectures and decoding methods, with particularly striking gains in challenging scenarios. Under nucleus sampling, MiniGPT-4's performance improves substantially from 845.30 to 1069.37, while LLAVA-1.5 maintains strong performance with significant improvement from 1625.22 to 1738.56. These comprehensive gains suggest that VISTA's approach to maintaining visual grounding enhances fundamental visual-language integration mechanisms, leading to better overall model behavior.

### 3.4. On Solving Gradual Visual Information Loss

Our analysis in Sec. 2.2 identifies gradual visual information loss as a substantial challenge for long sequence generation. To validate VISTA's effectiveness in addressing this issue, we compare token logits rankings w/ and w/o VISTA throughout generation. According to Fig. 5, VISTA not only improves the average ranking of hidden genuine tokens throughout generation but also reverses the concerning trend of hallucinated tokens, reducing their prominence in mid and late stages where hallucination typically occurs. Not surprisingly, VISTA also maintains a high ranking for decoded genuine tokens since VSV captures all visual clues of an image and reinforces them at all time steps. This quantitative evidence directly demonstrates VISTA's capability in maintaining visual grounding throughout the generation process. (see Appendix A.4 for additional results).

**Case Study.** Qualitative analysis further supports our quantitative findings above. We kindly refer readers to Appendix C for a collection of qualitative examples.

*Table 4.* Impact of window size on SLA performance. Layer ranges (X-31) indicate the span of layers used for logit augmentation, where X varies from 27 to 31. $C_S$ and $C_I$ denote CHAIR$_S$ and CHAIR$_I$ metrics, respectively.

| $\gamma$ | 31-31 | | | 30-31 | | | 29-31 | | | 28-31 | | | 27-31 | | |
|---|---|---|---|---|---|---|---|---|---|---|---|---|---|---|---|
| | $C_S$ | $C_I$ | F1 | $C_S$ | $C_I$ | F1 | $C_S$ | $C_I$ | F1 | $C_S$ | $C_I$ | F1 | $C_S$ | $C_I$ | F1 |
| 0.1 | 48.2 | 13.9 | 76.2 | 48.6 | 12.7 | 77.7 | 46.8 | 12.6 | 77.4 | 46.2 | 12.2 | 74.4 | 45.8 | 11.3 | 77.6 |
| 0.2 | 56.6 | 16.4 | 75.3 | 49.4 | 14.4 | 76.5 | 47.4 | 12.7 | 77.3 | 46.8 | 12.1 | 77.7 | 43.2 | 11.7 | 77.6 |
| 0.3 | 62.0 | 18.8 | 72.9 | 55.4 | 15.7 | 75.9 | 49.2 | 14.2 | 76.5 | 45.8 | 12.4 | 77.9 | **42.8** | **11.3** | **78.4** |
| 0.4 | 61.2 | 18.2 | 73.3 | 57.6 | 15.7 | 75.3 | 52.6 | 14.5 | 76.1 | 48.8 | 13.5 | 77.0 | 46.6 | 12.3 | 77.2 |

### 3.5. Ablation Study

To thoroughly investigate the effectiveness of VISTA, we gauge the practical latency of VISTA, and analyze how different VSV strength ($\lambda$) and SLA mixing ratio ($\gamma$) affect the model's performance in terms of hallucination reduction (CHAIR-S and CHAIR-I metrics) and overall quality (F1 score). Results are in Fig. 6. Additional ablation results are deferred to Appendix B.2.

*Table 5.* Measure of throughput and latency on LLAVA-1.5. Greedy decoding strategy is applied and listed as baseline.

| Methods | Greedy | VCD | PAI | **VISTA** (ours) |
|---|---|---|---|---|
| Latency (ms/token) ↓ | 28.54 (×1.0) | 58.34 (×2.04) | 57.78 (×2.02) | **36.32** (×1.27) |
| Throughput (token/s) ↑ | 35.04 (×1.0) | 17.14 (×0.49) | 17.31 (×0.49) | **27.53** (×0.79) |

**Efficiency.** The greedy decoding latency in Table 5 shows that VISTA is more efficient than the other test-time intervention strategy PAI and contrastive decoding strategy like VCD.

**VSV Strength** ($\lambda$). As we vary injection strength $\lambda$ from 0.0 to 0.18, both CHAIR-S and CHAIR-I scores improve significantly. However, using inappropriate scale (see Fig. 6) will cause clear degradation in F1, suggesting overemphasis on visual features can harm generation quality.

**SLA Mixing Ratio** ($\gamma$). The impact of $\gamma$ is studied across values from 0 to 0.4, revealing that moderate values of $\gamma$ (0.2-0.3) yield the best balance between hallucination reduction and generation quality, and higher $\gamma$ values ($\leq 0.4$) leads to degraded performance, hinting on the importance of syntactic information.

**Synergy & Robustness.** As visualized in Fig. 6, there exists a synergistic relation between $\lambda$ and $\gamma$. Moderate values of both parameters consistently outperform extreme settings of either component alone. Notably, generation

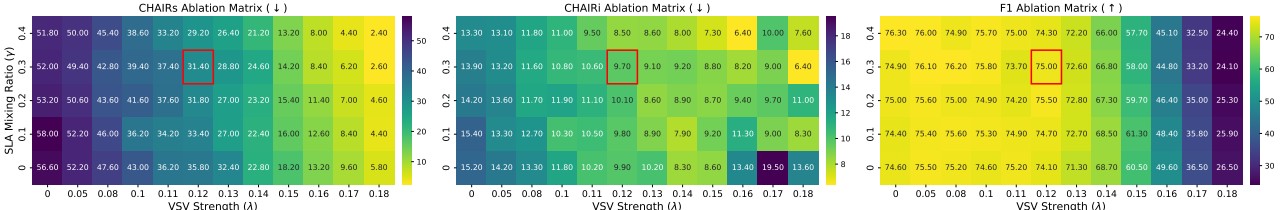

*Figure 6.* Ablation matrices for VSV strength ($\lambda$) and SLA mixing ratio ($\gamma$) on Shikra. Brighter color signifies the better performance. Red boxes highlight the parameter combinations we used. F1 score is included to demonstrate the overall generation quality.

quality remains stable (F1>72.0) across a broad range of parameter combinations, demonstrating the robustness of our approach. This complementarity suggests that VSV and SLA address different aspects of the hallucination problem.

**Window Size in SLA** ($w$). In Table 4, we explore window size from one layer up to five layers across different mixing ratios. As shown, larger window size generally reduces hallucination, and the optimal configuration achieves both strong hallucination reduction and the highest F1 score (78.4). Interestingly, we observe an inverse relationship between window size and optimal $\gamma$ values, suggesting that while broader layer spans capture richer visual information, they require more conservative mixing ratios to maintain generation stability.

## 4. Related Work

**Hallucination Mitigation in LVLMs.** Hallucination – the generation of content that is irrelevant, factually incorrect, or inconsistent with visual inputs (Bai et al., 2024) – represents a fundamental challenge in LVLM development. Research has identified three primary sources: limitations in visual encoder capabilities (Tong et al., 2024; Liu et al., 2024b; Shi et al., 2024), excessive reliance on learned parametric knowledge (Li et al., 2023b; Zhou et al., 2023; Leng et al., 2024; Huang et al., 2024), and noisy training data (Liu et al., 2023; Yu et al., 2024). Mitigation approaches span training-based solutions with refined datasets (Yue et al., 2024; Jiang et al., 2024), post-processing techniques including revision (Yin et al., 2023; Zhou et al., 2023) and verification (Chen et al., 2024; Sun et al., 2023), and inference-time interventions like Visual Contrastive Decoding (Leng et al., 2024) and enhanced attention methods (Liu et al., 2024g). Recent studies revealing "text inertia" (Liu et al., 2024g), where models generate similar hallucinations without visual input, highlight concerning reliance on learned text patterns. Similar to VISTA, VTI (Liu et al., 2024f) also applies representation engineering technique to LVLM hallucination domain, yet with a focus on the instability issue of visual representations. While prior findings advance our understanding, how hallucination propagates through model architectures remains elusive, and existing solutions often require external supervision or are hinged with specific decoding strategies.

**Contrastive Decoding in LVLMs.** Contrastive decod-

ing, originally introduced in NLP (Li et al., 2022; Shi et al., 2023), has emerged as a promising approach for reducing hallucination in LVLMs. Recent adaptations of this technique have explored various contrasting strategies: VCD (Leng et al., 2024) introduces visual-specific contrasts by crafting noisy visual tokens as negative samples, while DoLa (Chuang et al., 2024) innovates by contrasting logits distributions from different layers within the same model, using divergence measurements to dynamically select contrasting layers. Taking a temporal perspective, M3ID (Favero et al., 2024) proposes a "horizontal" strategy that contrasts current logits with those from previous timesteps. Other approaches extend contrastive techniques to attention mechanisms (Woo et al., 2024). While these methods primarily operate in the logits space, our VISTA takes a different approach by performing contrasts in the activation space and intervening at residual streams. This earlier-stage intervention strategy offers an efficient alternative that can complement existing decoding methods.

## 5. Conclusion and Limitations

This study investigates the hidden life of tokens in Large Vision Language Models (LVLMs) and introduces **VISTA** (**V**isual **I**nformation **S**teering with **T**oken-logit **A**ugmentation), a lightweight approach to mitigate hallucination. Through systematic analysis, we reveal that visual information gradually attenuates during text generation, but can be effectively restored through our framework's visual information steering and strategic use of early-layer logits. Extensive experimentation across diverse architectures and decoding strategies demonstrates that our framework significantly reduces hallucination while preserving generation quality. These findings not only illuminate the hidden dynamics of LVLM behavior but also establish visual information steering as a promising direction for enhancing the reliability of multimodal AI systems.

**Limitations.** VISTA is subject to several limitations. First, while VISTA demonstrates robustness across a range of hyperparameter values, optimal settings may vary across different architectures. Second, the effectiveness of VSV relies on the quality of visual cues extracted by the LVLM's vision encoder – models with weak visual encoding capabilities may see reduced benefits. Third, the current implementation

focuses on addressing hallucination in single-round tasks; adaptation to interactive scenarios like visual dialogue may require additional considerations.

## Impact Statement

This research advances methods for making large vision-language models more trustworthy and reliable through mitigating hallucination. While the proposed method demonstrates promising results, its effectiveness is subject to the inherent capability of large vision-language model, and improper usage may adversely affect model's performance. To the best of our knowledge, there are no ethical or other concerns that need to be addressed.

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

For each task, include objects, object properties (e.g., color, count, position), and relationships between objects. You must answer each content with a single word, separating different contents by commas. If no contents apply, write "None". Make sure there is no overlapping words between three tasks.

Answer 1: [Missing contents]
Answer 2: [Hallucinated contents]
Answer 3: [Accurate contents]

---

*Table 6.* The prompt used for GPT-4o to identify genuine and hallucinated words.

## A. Implementation Details for Token Ranking Analysis

### A.1. Prompt for GPT-4o

We employ GPT-4o as our oracle model for identifying three categories of tokens: hidden genuine tokens, decoded genuine tokens, and hallucinated tokens. The precise prompting strategy used to elicit these classifications is detailed in Table 6.

### A.2. Additional Implementation Details

A detailed token analysis algorithm is provided in Algorithm 1.

**Token Processing.** As shown in Algorithm 1, we derive genuine and hallucinated tokens from their corresponding word-level classifications. In cases where a single word decomposes into multiple tokens under the model's tokenization scheme, we adopt the first token as a representative proxy for the entire word. This approach ensures consistent handling of multi-token words while maintaining analytical tractability.

**Ranking Aggregation Protocol.** When computing cross-stage token rankings (visualized in Fig. 1 left), we implement a focused aggregation strategy that considers only the final five layers of the model, deliberately excluding rankings from earlier layers. This methodological choice mitigates the inherent embedding disparity between the model's decoding layer and preceding layers. Since the LLM's decoding head is specifically trained on final-layer hidden states, the reliability of token rankings decreases with distance from this layer. Consequently, we restrict our analysis to a window of layers proximate to the final layer to ensure robust and meaningful ranking estimates.

### A.3. Advantages of Token Ranking Analysis

**Why Not Attention?** While attention matrices have been extensively studied in LVLM hallucination research to understand information flow patterns and inform mitigation strategies (Huang et al., 2024; Liu et al., 2024g), our token ranking methodology offers several distinct advantages. Previous work, such as PAI (Liu et al., 2024g), has attributed hallucination phenomena like "text inertia" to insufficient attention allocation to visual tokens. However, this interpretation is potentially confounded by the presence of "anchor tokens" (Wang et al., 2023; Huang et al., 2024) that aggregate and redistribute information across the network. The existence of these information hubs means that reduced attention weights on visual tokens, particularly in later layers where visual information can be accessed indirectly through anchor positions, may not necessarily indicate information loss.

Token ranking analysis, by contrast, provides more direct insights into the model's processing of visual information. Through explicit tracking at the token level, this approach enables quantitative measurement of visual information preservation throughout the generation process. The methodology reveals gradual visual information degradation patterns that might be obscured in attention-based analyses. Furthermore, token ranking analysis uncovers previously unobserved phenomena such as hidden genuine information and early excitation patterns, which are not readily distinguishable through attention analysis alone. These capabilities make token ranking analysis particularly well-suited for investigating the mechanisms underlying hallucination in LVLMs.

---

**Algorithm 1** Token Ranking Analysis Framework

---

**Require:** Image $I$, LVLM model $M$, Oracle model $O$
**Ensure:** Token ranking matrices $R$ for each category
 1: /* Generate description using LVLM */
 2: $D \leftarrow M(I)$
 3: /* Classify tokens using Oracle */
 4: $W_{dec}, W_{hid}, W_{hal} \leftarrow O(I, D)$ {decoded, hidden, hallucinated}
 5: $T_{dec}, T_{hid}, T_{hal} \leftarrow \text{Tokenize}(W_{dec}, W_{hid}, W_{hal})$
 6: **for** each token category $T_c$ in $\{T_{dec}, T_{hid}, T_{hal}\}$ **do**
 7:    **for** $t = 1$ to $|D|$ **do**
 8:      **for** $l = 1$ to $L$ **do**
 9:        $h_t^l \leftarrow \text{GetHiddenState}(M, l, t)$ {Eq. 3}
10:        $\text{logits} \leftarrow H(h_t^l)$ {Eq. 4}
11:        **for** each token $x$ in $T_c$ **do**
12:          $R_t^l(x) \leftarrow \text{rank}(\text{logits}, x)$ {per Eq. 4}
13:        **end for**
14:      **end for**
15:    **end for**
16: **end for**
17: /* Compute stage-wise aggregation */
18: **for** each stage $s$ in $\{\text{early}, \text{mid}, \text{late}\}$ **do**
19:    **for** each category $c$ in $\{\text{dec}, \text{hid}, \text{hal}\}$ **do**
20:      $T_s \leftarrow \text{GetTimeSteps}(s)$ {timesteps in stage $s$}
21:      $L_f \leftarrow \text{GetFinalLayers}()$ {final 5 layers}
22:      $\bar{R}_s^c \leftarrow \text{mean}(\{R_t^l(x) \mid t \in T_s, l \in L_f, x \in T_c\})$
23:    **end for**
24: **end for**
**output** $R$ {Ranking matrices for all categories}

---

**Limitations.** Our token ranking analysis approach also pose certain limitations:

1. **Oracle Model Reliability:** While GPT-4o serves as our oracle for identifying genuine and hallucinated content, this process can introduce potential biases and uncertainties. The oracle model's classifications may not perfectly align with human judgments, and its own biases could influence the categorization of tokens. This is particularly challenging for nuanced cases where the distinction between genuine and hallucinated content is subtle.

2. **Embedding Space Discrepancy:** Another limitation arises from applying the LLM's decoding head to hidden states from earlier layers. Since the decoding head is specifically trained on final-layer representations, there exists an embedding space misalignment when analyzing preceding layers. This discrepancy becomes more pronounced for layers distant from the final layer, potentially leading to less reliable token rankings in earlier stages of the network. While our analysis mitigates this by focusing on layers proximate to the last layer, the issue remains inherent to the methodology.

Despite above limitations, our analysis provides valuable insights into LVLM's behavior and has proven effective in motivating our hallucination mitigation approach.

### A.4. Additional Token Ranking Analysis

In the main text, we present token ranking analysis results for LLAVA-1.5. Here, we extend this analysis to other architectures to demonstrate the generalizability of our observations. As shown in Fig. 7, the cross-stage token ranking analysis on Shikra exhibits similar patterns to those observed in LLAVA-1.5, with genuine tokens experiencing gradual rank degradation while hallucinated tokens become increasingly prioritized across generation stages. The layer-wise analysis presented in Fig. 8 further corroborates the early excitation phenomenon, where semantic tokens achieve peak activation in layers preceding the

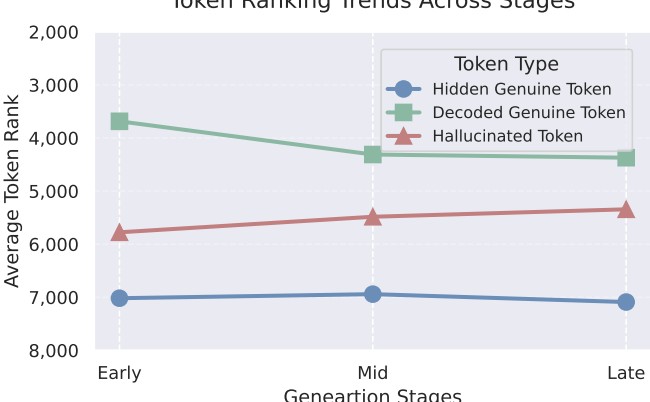

*Figure 7.* Cross-stage token ranking on Shikra.

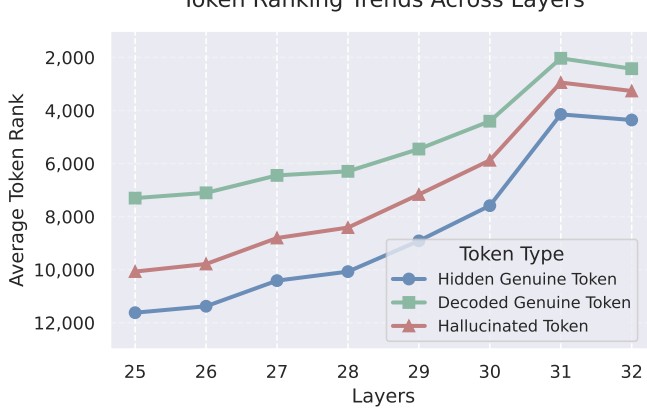

*Figure 8.* Layer-wise token rankings on Shikra.

final decoding layer. Fig. 9 presents a comparative analysis between vanilla decoding and VISTA (greedy-based) on Shikra. The results demonstrate VISTA's effectiveness in maintaining the ranking of genuine tokens throughout the generation process while simultaneously suppressing the promotion of hallucinated tokens. This pattern is consistent with our findings for LLAVA-1.5, suggesting that the phenomena we identified and the effectiveness of our mitigation strategy generalize across different LVLM architectures. The consistency of these patterns across architectures with distinct design choices (linear projector in Shikra versus Q-former in other models) provides strong evidence for the fundamental nature of these phenomena in LVLM generation dynamics.

## B. Additional Experiments

### B.1. MMHal-Bench Results For Other Decoding Strategies

We further report results of MMHal-Bench under beam search (Fig. 10) and nucleus sampling (Fig. 11). As demonstrated in figures, VISTA consistently improves overall performance across all evaluated LVLMs under both decoding strategies. The performance trends remain consistent with those observed under greedy decoding in the main text, further validating the robustness of our approach across different inference strategies. These comprehensive results demonstrate that VISTA's mechanisms for maintaining visual grounding and promoting semantic richness are effective regardless of the chosen decoding strategy.

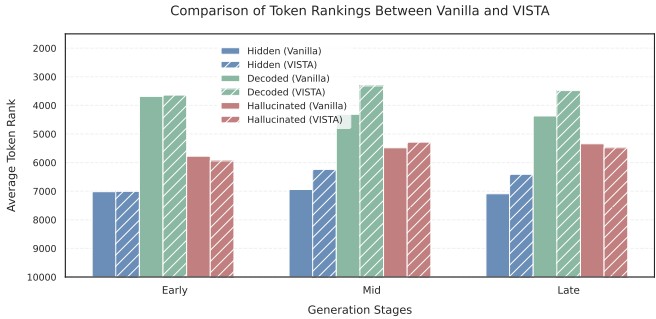

Figure 9. Cross-stage token ranking comparison between greedy and VISTA (greedy-based) on Shikra. VISTA effectively promotes the ranking of genuine tokens while depressing hallucination tokens.

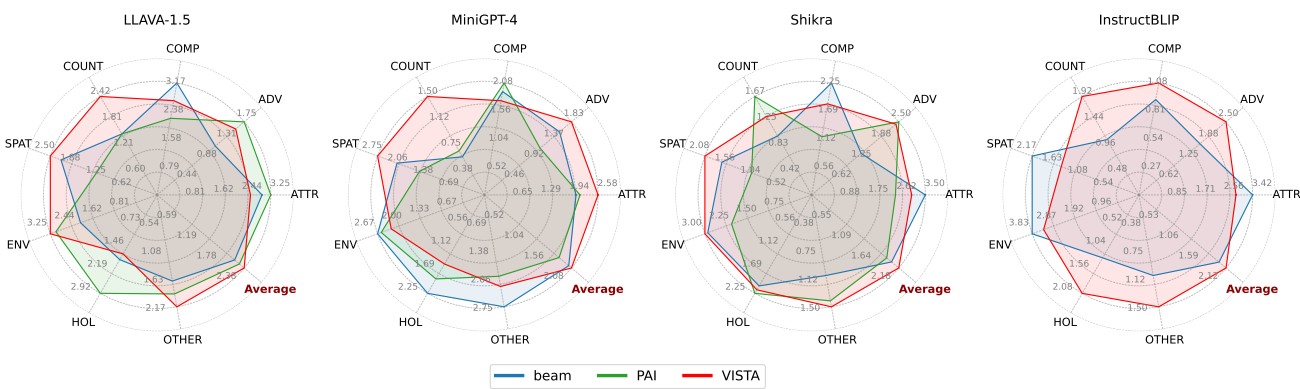

Figure 10. Performance comparison on MMHal-Bench using beam search.

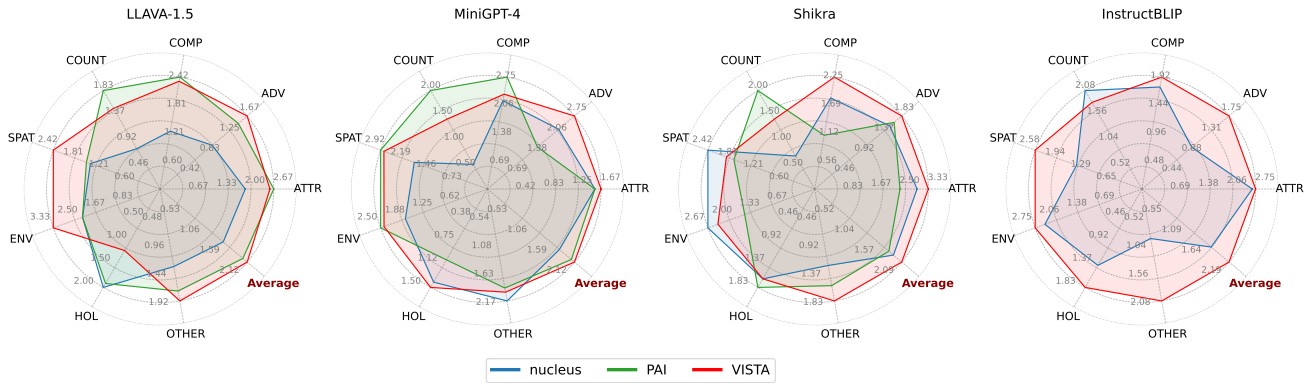

Figure 11. Performance comparison on MMHal-Bench using nucleus sampling.

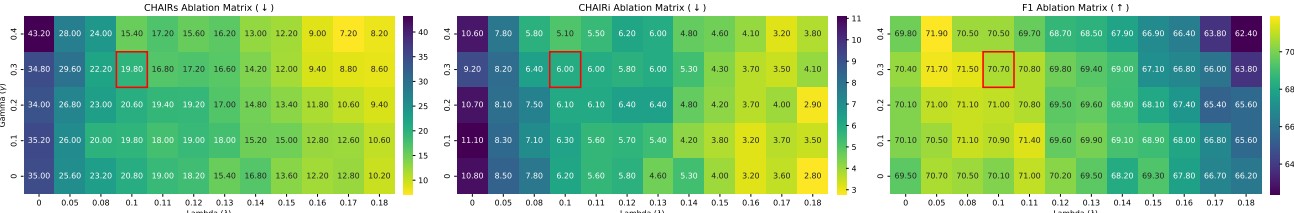

*Figure 12.* Ablation matrices for VSV strength (λ) and SLA mixing ratio (γ) on MiniGPT-4. Brighter color signifies the better performance, and red boxes highlight the parameter combinations used in Table 1. F1 score is included to indicate the overall generation quality.

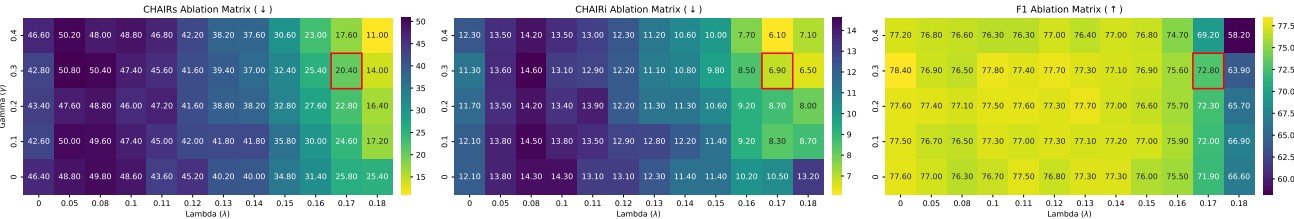

*Figure 13.* Ablation matrices for VSV strength (λ) and SLA mixing ratio (γ) on LLAVA-1.5. Brighter color signifies the better performance, and red boxes highlight the parameter combinations used in Table 1. F1 score is included to indicate the overall generation quality.

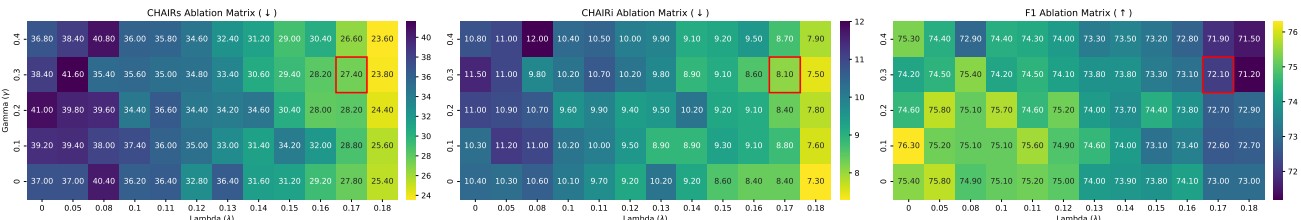

*Figure 14.* Ablation matrices for VSV strength (λ) and SLA mixing ratio (γ) on InstructBLIP. Brighter color signifies the better performance, and red boxes highlight the parameter combinations used in Table 1. F1 score is included to indicate the overall generation quality.

## B.2. Additional Ablation Results

In addition to the ablation studies presented in the main text, we further provide detailed ablation results on LLAVA-1.5, MiniGPT-4, and InstructBLIP to validate our hyperparameter choices. Figure 12 presents the ablation matrices for MiniGPT-4, examining the impact of VSV strength ($\lambda$) and SLA mixing ratio ($\gamma$) on CHAIR$_S$, CHAIR$_I$, and F1 scores. The results reveal similar trends to those observed in Shikra, though with a slightly lower optimal $\lambda$ value of 0.1, suggesting architecture-specific sensitivity to visual steering. For LLAVA-1.5 (Figure 13), the ablation matrices demonstrate particularly strong performance improvements with stronger VSV value ($\lambda = 0.17$). The InstructBLIP results (Figure 14) show robust performance across a broader range of parameter combinations, with optimal performance achieved at $\lambda = 0.17$ and $\gamma = 0.3$, matching the configuration used for LLAVA-1.5.

Across all architectures, we observe a consistent pattern where moderate values of both VSV strength and SLA mixing ratio yield the best balance between hallucination reduction and generation quality. This consistency, despite architectural differences, validates the generality of our approach while highlighting the need for architecture-specific fine-tuning of hyperparameters for optimal performance.

# C. Case Study

In this section, we extend our qualitative evaluation across all four architectures to demonstrate VISTA's effectiveness in reducing hallucination while promoting genuine information. Figures 15-18 present comparative examples between vanilla decoding and VISTA for LLAVA-1.5, MiniGPT-4, Shikra, and InstructBLIP respectively. For each example, hallucinated content is highlighted in **red**, while genuine information that was previously omitted or wrongly described but successfully recovered by VISTA is marked in **blue**. The examples demonstrate VISTA's consistent ability to reduce hallucination across different architectures while eliciting pertinent visual details that vanilla decoding fails to capture.

**Greedy:** The image captures a young boy in a baseball uniform, standing at home plate and holding a baseball bat, ready to swing. He is surrounded by a crowd of people, including teammates and spectators, who are watching the game intently. There are several chairs placed around the field, with some of them occupied by the spectators. **A baseball glove can be seen on the ground, likely belonging to one of the players.** The scene is lively and filled with anticipation as the young boy prepares to hit the ball.

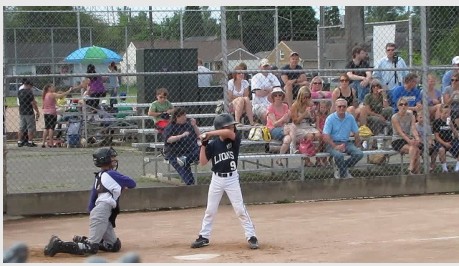

**VISTA:** The image depicts a baseball game in progress with a young boy standing at home plate, holding a baseball bat and preparing to hit the ball. The boy is wearing a baseball uniform and is the main focus of the scene. **There are several people in the background, some of them sitting on benches and others standing around the field. A few of the spectators are holding handbags, and a few are wearing hats.** The scene captures the excitement and anticipation of the game.

**Greedy:** The image features a red stop sign with a unique message written on it. The sign reads "Sook Sook Eee It Stop," which is a playful variation of the standard "Stop" message. The sign is placed on a pole, **and it is situated in front of a building**. The scene also includes a tree in the background, adding to the overall atmosphere of the location.

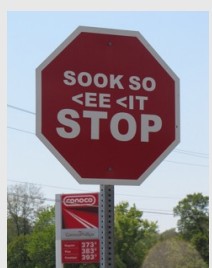

**VISTA:** The image features a red stop sign with a playful message written on it. The sign is positioned on a pole, and the message reads "Sook So So Stop." **The sign is located in a street setting, with a tree in the background.**

LLAVA-1.5

**Greedy:** The image features a red and yellow train traveling down the tracks, **with a large silver train car following behind it.** The train is moving along the tracks, **and there are several other trains visible in the background, indicating a busy train station. In addition to the trains, there are multiple people scattered throughout the scene, likely waiting for their respective trains or simply passing by. Some of these individuals are closer to the foreground, while others are further away, giving a sense of depth to the image.**

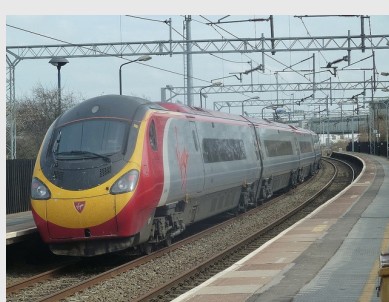

**VISTA:** The image features a train traveling on a train track. The train is a silver and red color, **with a yellow front**. The train is moving along the tracks, **and there are multiple train tracks visible in the scene.** The train is positioned in the middle of the scene, and the tracks are spread across the entire image.

**Greedy:** The image features a man wearing a helmet and riding a black motorcycle down a street. He is the main focus of the scene, with the motorcycle taking up a significant portion of the image. The man appears to be enjoying his ride, possibly commuting or traveling down the road. **In the background, there are a few other people visible, but they are not the main subject of the image. The street appears to be relatively empty, with no other vehicles or significant obstacles visible.**

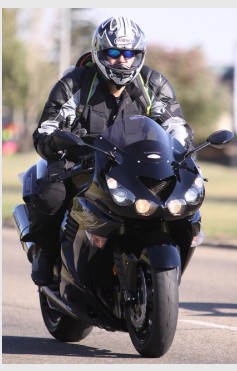

**VISTA:** The image features a person riding a motorcycle on a street. The person is wearing a helmet **and a backpack**, ensuring safety while riding. The motorcycle is black and occupies a significant portion of the scene. The person is positioned in the middle of the image, with the motorcycle taking up the majority of the frame.

*Figure 15.* Case study for LLAVA-1.5.

**Greedy:** The image shows a woman in a blue dress and white hat holding a tennis racket on a tennis court. **The woman is standing on the left side of the court** and the net is in the background. There are trees in the background and the sky is visible **in the top right corner of the image**. The woman is wearing white shoes and her hair is pulled back into a ponytail. The racket is **held with both hands** and the woman is ready to serve the ball.

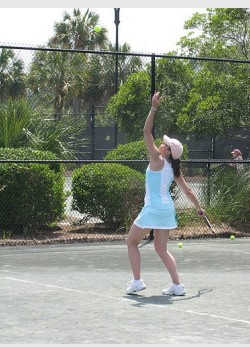

**VISTA:** The image shows a woman in a blue dress and white tennis shoes on a tennis court. **She is holding a tennis racket in her right hand** and is about to serve the ball. There are trees in the background.

**Greedy:** The image shows a person riding a green dirt bike in the air, with a cloudy sky in the background. The person is wearing **a black helmet, black gloves,** and a green jumpsuit. **The bike has a black frame and green accents**, and it is being ridden by a person in black boots and gloves. The dirt in the background is brown **and there are some trees in the distance.**

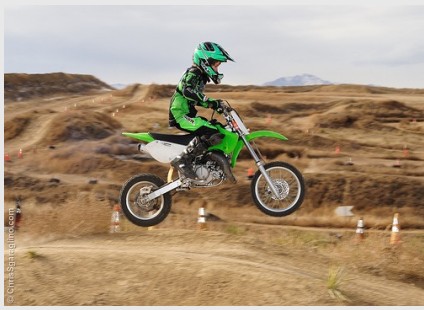

**VISTA:** The image shows a person on a green dirt bike in the air, riding over a dirt hill. The person is wearing a green helmet and a green jumpsuit. The background is a dirt hill **with a few small dirt mounds on it.**

**MiniGPT-4**

**Greedy:** The image shows a group of sheep standing in a muddy field. The sheep are all white **and appear to be of a similar age and size**. They are standing in a circle, with their heads facing inward **and their tails tied together**. In the center of the circle is a large pile of hay or straw, **which appears to have been stacked up by the sheep**. The sky is cloudy and there are trees in the background.

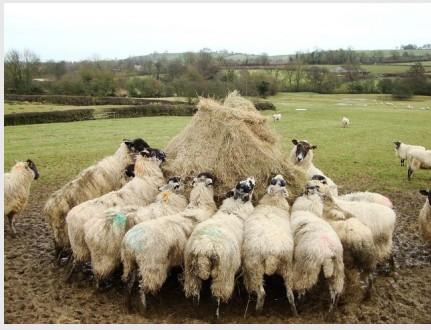

**VISTA:** The image shows a group of sheep standing in a muddy field. The sheep are all white and appear to be in a group. There is a pile of hay in the foreground, and in the background there are trees and a blue sky. **The sheep appear to be in the process of eating the hay.**

**Greedy:** The image shows a man sitting on the hood of a car with his feet on the ground. He is wearing a white shirt and blue jeans and is talking on a cell phone. **In the background, there is a dark sky and some trees.** The car is a black sedan **with its hood open.**

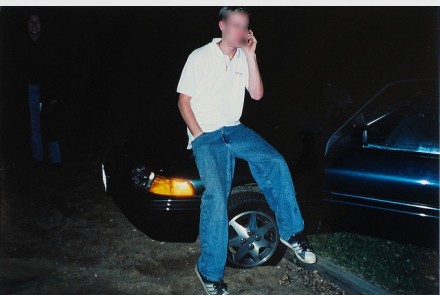

**VISTA:** The image shows a man sitting on a car in a dark area. He is wearing a white shirt and blue jeans. The car in the background is a black car. The man is holding a cell phone to his ear. **The image is taken at night.**

*Figure 16.* Case study for MiniGPT-4.

**Greedy:** The image features two young girls sitting together at a dining table, enjoying their donuts. One girl is on the left side of the table, while the other is on the right. They both appear to be quite content with their treats. **There are two chairs in the scene, with one being occupied by the girl on the left and the other chair by the girl on the right.** The table they are seated at is quite long, extending across the image. **There is also a cup placed on the table near the right side.**

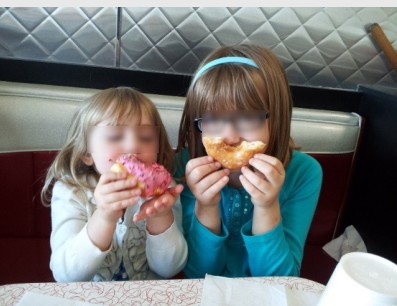

**VISTA:** The image features two young girls, **one with glasses**, enjoying donuts at a table. They are both holding their respective donuts, with one girl on the left and the other on the right. The girls are positioned in the middle of the scene, and it appears they are also sitting on a chair. There is a chair visible in the background, but it is not the main focus of the scene. The overall atmosphere of the image is that of enjoying a shared treat together.

**Greedy:** The image features the iconic Big Ben clock tower in London, with a cloudy sky overhead. The tower is part of the Houses of Parliament and stands strong against the backdrop. There is a large tree in the foreground, partially obscuring the view of the clock. **Several birds can be seen flying near the clock tower, with one bird perched at the top left corner and two more birds flying closer to the bottom left of the image. The birds' presence adds a lively touch to the scene.**

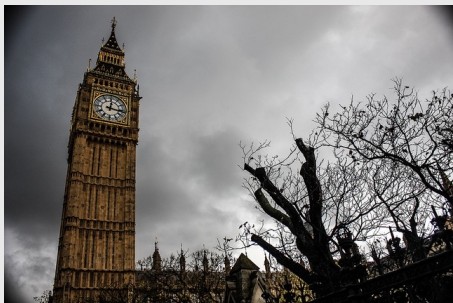

**VISTA:** The image features a cloudy sky, a towering building, and a large clock on the side of the building. The clock is visible, along with the tower's overall size, making it a prominent feature of the scene. **The sky appears to be a mix of various shades, including gray, beige, and even slightly blue,** creating a picturesque setting for the scene.

**Shikra**

**Greedy:** The image features a man wearing a black shirt and a purple tie. He is standing in a room with a wooden floor and a painting on the wall. The man appears to be wearing glasses, which adds to his distinguished look**. In the background, there are two TVs visible. One is placed on the left side of the room, while the other is located closer to the center. Additionally, there are two books in the scene, one on the floor near the man and the other further to the right. The overall setting suggests a well-decorated living space.**

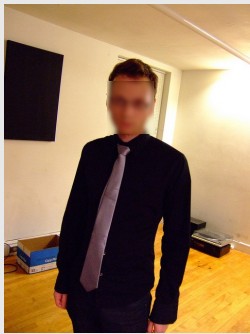

**VISTA:** The image shows a young man wearing a black shirt and a purple tie. **He is standing confidently, with a slight smile on his face**. The man appears to be wearing glasses, which are tinted black. He is positioned in the middle of the frame, and **there is a box nearby**. The man's tie is purple, and **it is tucked under his shirt**.

**Greedy:** The image captures a man skiing down a snow-covered slope on a mountain. He is wearing an orange helmet and is skiing near a cluster of trees. The skier appears to be enjoying the downhill run as he moves quickly through the snow. **In addition to the skier, there are two backpacks visible in the scene. One is located near the left side of the skier, while the other is positioned closer to the middle of the image. The presence of backpacks suggests that the skier might be carrying some gear or supplies for his skiing adventure.**

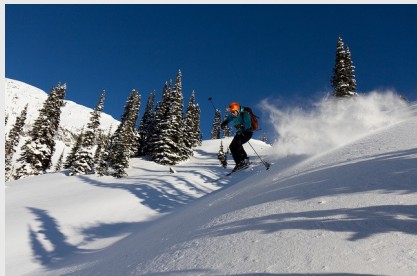

**VISTA:** The image features a snowy mountain slope bustling with activity. There is a skier, specifically, who is skiing down the snow-covered slope. The skier is wearing an orange helmet and is also riding on skis, making their way through the snowy landscape. **The skier's ski poles can be seen in their hand while skiing**. The scene also highlights the presence of trees surrounding the skiing area, adding to the picturesque quality of the scene. The trees add to the lively atmosphere of the area, making this a great spot for skiing and exploring.

*Figure 17.* Case study for Shikra.

**Greedy:** The image features a dog **and a cat** playing in the ocean, with the dog splashing in the water **and the cat watching from a distance**. **The dog is positioned closer to the water, while the cat is further away, possibly observing the dog's playful behavior.** The water is foamy and bubbly, indicating a lively scene of the animals enjoying the ocean.

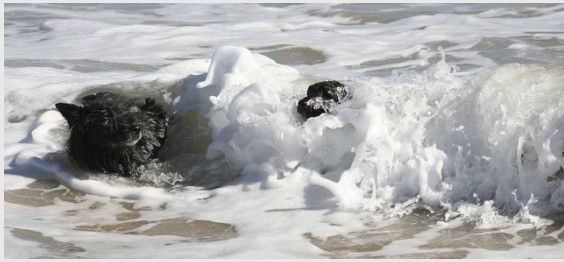

**VISTA:** **Two dogs are playing in the ocean, with one dog splashing in the water and the other dog enjoying the waves.** The dogs are in the ocean, splashing and playing in the waves, with one dog in the foreground and the other in the background. The dogs are having a fun time in the ocean, enjoying the water and the playful waves.

**Greedy:** The image features a man standing on a sandy beach, flying a kite in the sky. He is wearing a red shirt and is holding the kite string, which is stretched out in the air. The kite is flying high above the man, creating a beautiful sight against the clear blue sky. The beach is surrounded by water, with a body of water visible in the background. **There are several people in the scene, including a woman and a child, who are also enjoying the beach and the kite flying.**

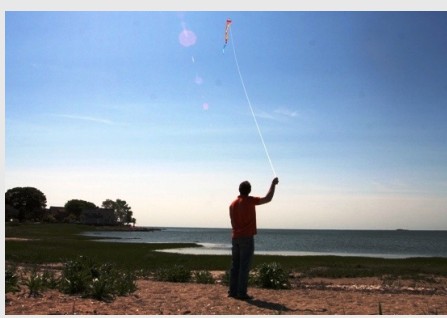

**VISTA:** A man is standing on a beach, flying a kite in the sky. He is wearing a red shirt and is focused on flying the kite, which is high up in the sky. The man is in a relaxed and enjoyable setting, enjoying the **sunny day** and the serene atmosphere of the beach. **The kite is a colorful** and playful addition to the scene, adding to the overall sense of fun and leisure.

**InstructBLIP**

**Greedy:** The image features a person standing in the snow, holding a snowboard. The snowboard is positioned on the ground next to the person, who is wearing a **blue jacket and pants.** There are several snowboards visible in the scene, with one placed on the ground **and another on a snowboard rack. The snowboard rack is located on the left side of the image, with a few snowboards placed on it. The snowboard rack is likely used to store and organize the snowboards for easy access during snowboarding activities.**

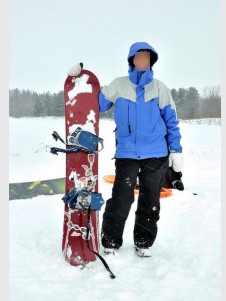

**VISTA:** A woman is standing in the snow, wearing a blue and white jacket, and **holding a snowboard in her hands. She is posing for a photo in a winter setting, likely in a snowy field or on a ski slope**. There are several other snowboards visible in the scene, likely belonging to other people or in the background. **The snowboard in the foreground is a red and black snowboard, and the woman is smiling as she poses for the picture.**

**Greedy:** The image features a brown horse standing in a grassy field, with a white bird perched on its back. The bird is positioned on the horse's back, **with its wings spread out, and its body resting on the horse's neck**. The horse and the bird are both standing in the field, with the bird's head slightly turned towards the left side of the image. The grassy field is filled with various plants and shrubs, creating a lush and natural setting.

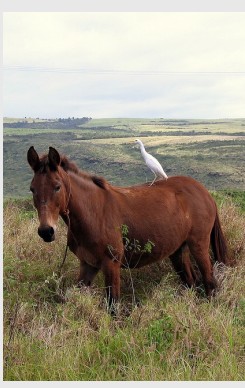

**VISTA:** There is a brown horse standing in a grassy field, with a white bird perched on its back. The horse and bird are both in the foreground of the scene, **with the bird resting on the horse's back**. The horse is positioned in the middle of the field, with the bird on its back, and **the background features a rolling landscape with hills and a distant horizon**. The scene depicts a peaceful moment in a natural setting, with the horse and bird enjoying a relaxing moment in the grassy field.

*Figure 18.* Case study for InstructBLIP.

