# OpenReview forum: "The Hidden Life of Tokens: Reducing Hallucination of Large Vision-Language Models Via Visual Information Steering"
_ICML.cc/2025/Conference — ICML 2025 poster_

### Official Review · Reviewer_ypPL · 2025-03-12

**Overall Recommendation:** 3

**Summary:**

This paper introduces a inference-time framework called VISTA that aims to reduce hallucinations in LVLMs. The authors conduct a detailed study of token “logit rankings” across both layer depth and the temporal dimension in a generated sequence. Their analysis reveals three observations: (1) Gradual loss of visual grounding over the course of generation, (2) Early excitation of semantically meaningful tokens in penultimate or nearby layers, and (3) Hidden genuine information that the model “knows” but fails to include in its output. Building on these insights, they propose VISTA, which combines two complementary techniques: (1) Visual Steering Vector (VSV): a direction in the model’s activation space, computed by contrasting the residual streams obtained from the “positive” prompt containing the image tokens vs. a “negative” prompt that omits them. Injecting this vector at each decoding step enforces better retention of visual information. (2) Self-Logits Augmentation (SLA) aggregates token logits from earlier (penultimate) layers, where grounded or semantically rich tokens peak, and then mixes them with the final layer’s logits for next-token sampling. Experiments on four LVLM architectures (LLaVA-1.5, MiniGPT-4, Shikra, InstructBLIP) and three standard decoding regimes (greedy, beam, nucleus sampling) demonstrate that VISTA consistently outperforms baselines on several hallucination-specific benchmarks (CHAIR, POPE, MMHal) and general capability tests (MME).

**Claims And Evidence:**

1. Hallucination emerges partly because the model’s language priors overshadow the diminishing visual signals, especially in later decoding steps.
- The token-rank analysis across layers/time shows that visually grounded tokens steadily drop in probability ranks, while hallucinated tokens rise. Figures 1-2 illustrate this trend clearly.

2. Meaningful tokens reach their maximal activation in intermediate layers (e.g., the penultimate), not the final layer – which is more biased toward function words.
- Again, the “logit lens” technique indicates that object/attribute tokens see higher probabilities earlier than the final layer, which frequently emphasizes grammatical tokens.

3. VISTA mitigates hallucinations by reinforcing the missing or eroded visual cues (via VSV) and leveraging earlier layer logits (via SLA).
- VISTA yields consistent gains in object hallucination metrics (CHAIRS, CHAIRI) of ∼40% relative improvement across multiple models and decoding algorithms. They also show smaller improvements in broad tasks (MME) and short-answer tasks (POPE), indicating that the method can generalize.

**Essential References Not Discussed:**

Might not mention older works on “residual stream editing” or “concept neurons.” Possibly not critical but could be relevant.

**Experimental Designs Or Analyses:**

Solid coverage of standard vision–language tasks and ablation on the synergy of the two main modules. However, the approach might be somewhat complicated in real usage: “steering vectors” are computed for each image.

**Methods And Evaluation Criteria:**

They compute a “steering vector” from contrast (with/without image tokens) and inject it in the residual streams, plus a partial ensemble of earlier layer logits to finalize the token distribution. They measure CHAIR, POPE, etc and validated against strong existing baselines like VCD, OPERA, PAI, and DoLa.

**Other Comments Or Suggestions:**

Read above and below.

**Other Strengths And Weaknesses:**

Strengths:
1. Intriguing token ranking analysis that reveals new insights. Thorough token-level analysis using logit lens clarifies when/how hallucinated vs. grounded tokens appear.
2. Strong performance gains across multiple models, decoding methods, and tasks.

Weaknesses:
1. Relying on the vision encoder’s quality. If the base model’s image embeddings are poor or incomplete, the “steering vector” might not help.
2. The method can require hyperparameter tuning for different models. For example, the injection scale hyperparameter might be sensitive; not thoroughly studied.
3. The VSV computation can be somewhat heavy if done for each image. It needs thorough time analysis.

**Questions For Authors:**

1. How does VISTA perform if there are many more entity mentions in an image than the user explicitly queries about? Could it lead to “overstuffing” of visual details?
2. If the system has to maintain visual grounding over multiple user queries, do we re-inject the same VSV at each turn, or might some turn-based adaptation be needed?
3. Do you see major differences if the image is extremely cluttered? Do you rely on the visual token embeddings to be relatively stable?
4. Could early-layer steering alone fix the problem without discretization or advanced overshadowing?

**Relation To Broader Scientific Literature:**

They cite relevant VLM hallucination works. The references are okay but possibly missing a deeper link to linear prompt engineering or “prefix tuning” for controlling generation.

**Theoretical Claims:**

This work is primarily empirical and methodological. The claims are mostly validated through data and are consistent with known ideas on how Transformers store information in intermediate hidden states. There are no deep new formal proofs, but no obvious theoretical errors either.

---

> ### Author Rebuttal · Authors · 2025-03-29
>
> *We cordially appreciate your careful review and insightful questions. We are thrilled that you find our analysis __"reveals new insights"__., our method gains __"strong performance"__ and __"can generalize"__. See below for detailed replies.*
>
> **Q1. Regarding vision encoder’s quality**
> >VISTA focuses on hallucinations caused by parametric priors and `does not set extra requirements for visual encoder's quality`, as evidenced by its effectiveness across four widely adopted architectures. The quality requirement applies to standard public architectures, `consistent with existing methods` such as VCD, PAI, and OPERA. VISTA complements ongoing research focused on improving vision encoders.
>
> **Q2. Regarding hyperparameter tuning**
> >There are two hyperparameters: intervention strength $\lambda$ and mixing ratio $\gamma$.
> > * For $\gamma$, we found $\gamma\approx0.3$ `consistently effective across all four architectures`.
> > * Optimal $\lambda$ can vary by model; however, a broad range of $\lambda$ values are effectively for each model, `as analyzed in the Synergy & Robustness subsection (p.7, line 381)` and shown in `Figs. 6, 12–14`, demonstrating VISTA's robustness across different $\lambda$ configurations.
>
> **Q3. Regarding computational overhead**
> >VISTA is an efficient de-hallucination strategy that `introduces minimal computational overhead` and demonstrates `better efficiency than baseline approaches.` We clarify this in below:
> >* **For captioning, summarization, and open-ended generation tasks**: The textual prompt is static (e.g., "please describe the image..."). VISTA forwards the textual prompt `only once` w/o image to cache activations (minimal overhead). For positive case, VISTA acts alike vanilla inference which forwards the visual+textual tokens. The only difference is before generating new tokens, VISTA builds VSV via `simple vector arithmetic` (marginal overhead), and proceeds with intervention. VISTA `does not forward twice` the input tokens, and it can use KV cache of them during generation` (overhead of this is also small).
> >* **For QA tasks**: Textual prompts vary per query, potentially requiring an additional forward pass of textual tokens. However, textual tokens typically account for `less than 10% of total prompt length` compared to image tokens`. Thus, the additional computational overhead remains mild relative to vanilla inference.
> >* **Empirically**, VISTA demonstrates `better efficiency` compared to other comparable methods (see `Table 5`). We kindly refer the reviewer to Reviewer 5sMs (Q2) for an extended efficiency comparison.
>
> **Q4. Is there "overstuffing" of visual details?**
> >* We'd like to first clarify a potential caveat. The visual steering vector (VSV) does not merely amplify visual details. The positive vector $V_p$ is conditioned `jointly on visual and textual tokens, capturing relevant visual-textual relationships` via attention heads, while the negative vector ($V_n$) reduces textual-only priors. Consequently, VSV `does not overstuff visual details` that might blur the query's focus.
> >* This is empirically validated on MMHal-Bench (Fig.4), where `each query specifically targets an entity or relation within visually complex scenes` where many other entities and relations are existing, and VISTA `performs well`.
>
> **Q5. Multi-query scenario**
> >As clarified in Q4, VSV is constructed per visual-query pair and is expected be recalculated per new query. However, as analyzed in Q3, constructing VSV incurs only mild computational cost due to KV caching and the relatively short length of query tokens compared to visual/system tokens. As a result, it is still efficient to use VISTA under multi-query scenarios.
>
> **Q6. Regarding extremely cluttered images**
> >Inspired by your insightful question, we conducted a new experiment where 500 images from MSCOCO are randomly selected and divided into two groups (heavy and light) according to the degree of cluttering (GPT-4o is used to rate a cluttering score for each image). Results below (on LLAVA-1.5) demonstrate that `heavily cluttered images is inherently more challenging`; yet VISTA `significantly outperforms vanilla decoding` in both scenarios.
>
> >|Cluttering degree|$C_S$↓/$C_I$↓|
> |:-|:-|
> |Heavy (greedy)|52.0 / 13.0|
> |Heavy (ours)|**26.5 / 5.3**|
> |Light (greedy)|40.5 / 11.7|
> |Light (ours)|**22.0 / 5.6**|
>
> **Q7. Could early-layer steering fix the problem?**
> >Following your suggestion, we conduct additional experiment applying early-layer steering (first 15 layers) for heavy and light cluttered images. As shown below, `steering across all layers consistently outperforms early steering`. Interestingly, the `performance gap between early steering and steering all layers is smaller for lightly cluttered images`, suggesting early layers handle much of the visual processing for easy cases.
>
> >|Cluttering degree|$C_S$↓/$C_I$↓|
> |:-|:-|
> |Heavy (early)|37.0 / 10.2|
> |Heavy (all)|**26.5 / 5.3**|
> |Light (early)|26.0 / 7.1|
> |Light (all)|**22.0 / 5.6**|

---

### Official Review · Reviewer_5sMs · 2025-03-13

**Overall Recommendation:** 3

**Summary:**

Through the observation of the LVLM generation process, this paper introduces a hallucination mitigation method, VISTA, which includes a visual steering vector and logit ensemble. Experiments demonstrate that it outperforms existing methods.

## Update after rebuttal
I agree with the authors' rationale and will maintain my initial score (weak accept).

**Claims And Evidence:**

Yes.

**Essential References Not Discussed:**

All related works are well discussed.

**Experimental Designs Or Analyses:**

Yes.

**Methods And Evaluation Criteria:**

Yes.

**Other Comments Or Suggestions:**

See the weakness section.

**Other Strengths And Weaknesses:**

Strengths
- Applying an effective method to each problem: a steering vector to enhance the model's focus on visual information and a logit ensemble to address the early excitation of semantically meaningful tokens.

Weaknesses
- Lack of novelty:
-  As the authors mentioned, each observation has already been discussed in previous literature. What is being observed for the first time in this paper? Is it the identification of three types of tokens and the observation of the LVLM generation process from that perspective?
- Also, the logit ensemble method has already been proposed in recent works [1, 2]. Please verify its superiority compared to those works.

[1] Do You Keep an Eye on What I Ask? Mitigating Multimodal Hallucination via Attention-Guided Ensemble Decoding, ICLR 2025

[2] Self-Introspective Decoding: Alleviating Hallucinations for Large Vision-Language Models, ICLR 2025

**Questions For Authors:**

1. The design of the visual steering vector. What if $V_n$ is directly subtracted? What would the results be?

**Relation To Broader Scientific Literature:**

The key contributions relate to MLLM hallucination mitigation.

**Theoretical Claims:**

N/A

---

> ### Author Rebuttal · Authors · 2025-03-27
>
> *We sincerely appreciate your detailed review and constructive suggestions. We are encouraged to find our framework is being recognized as __"Applying an effective method to each problem"__. All __claims are evidenced__, and related work are __"well discussed"__.  Below we address your questions in detail*.
>
> **Q1. Regarding novelty**
>
> >Our VISTA is `novel` in terms of both its `analysis` and `methodology`. To our best knowledge:
>
> >* VISTA's token ranking analysis is `the first work` inspecting `internal token-level dynamics` of LVLMs.
> >* VISTA systematically uncovers `three novel observations about how LVLMs store and process visual information`: (1) gradual visual information loss, (2) early excitation, and (3) hidden genuine information.
> >* VISTA resembles `the first activation-space intervention method for reducing LVLM's hallucination`, effectively addressing observed phenomena.
>
> >As suggested in your second bullet, VISTA indeed `provides a new perspective` by tracking the rankings of proposed "hallucinated", "genuine", and "hidden genuine" tokens. As a result, VISTA unveils three novel observations, and contributing valuable new insights to the field (supported by Reviewer ypPL).
>
> **Q2. Comparison and Discussion with [1] [2]**
>
> >Below, we first compare VISTA with [1] and [2] in terms of performance and efficiency, followed by an in-depth discussion of the differences between VISTA and these methods. *The extended comparison and discussion will be included in the next revision.*
>
> * **Performance**
>
> >|CHAIR|LLAVA-1.5| MiniGPT-4|Shikra|InstructBLIP|
> |:-|:-|:-|:-|:-|
> ||$C_S$/$C_I$↓|$C_S$/$C_I$↓|$C_S$/$C_I$↓|$C_S$/$C_I$↓|
> |ED [1]|43.0/14.0|-|-|-|
> |CT$^2$S (greedy) [2]|44.2/12.2|-|44.8/12.8|42.3/12.4|
> |__Ours__ (greedy)|__20.4/6.9__|__19.8/6.0__|__31.4/9.7__|__27.4/8.1__|
> |CT$^2$S (sampling) [2]|45.0/11.7|-|46.0/12.9|43.6/13.1|
> |__Ours__ (sampling)|__24.0/8.2__|__18.4/6.4__|__31.8/9.7__|__29.4/9.1__|
>
> >|POPE (avg)|LLAVA-1.5| MiniGPT-4|Shikra|InstructBLIP|
> |:-|:-|:-|:-|:-|
> ||Acc/F1↑|Acc/F1↑|Acc/F1↑|Acc/F1↑|
> |ED [1]|__86.31__/85.86|-|-|-|
> |CT$^2$S (greedy) [2]|85.94/85.92|-|81.84/82.23|82.30/83.58|
> |__Ours__ (greedy)|86.15/__86.29__|__75.96/77.11__|__82.44/82.47__|__84.87/84.95__|
> |CT$^2$S (sampling) [2]|85.14/85.41|-|80.66/__81.65__|81.49/82.70|
> |__Ours__ (sampling)|__85.35/85.54__|__66.96/68.05__|__81.01__/81.15|__83.11/83.27__|
>
> >As shown above, VISTA achieves `superior hallucination reduction performance`, particularly in long-sequence generation tasks such as CHAIR.
>
> * **Efficiency**
>
> >We further compare efficiency among methods. Given different evaluation criteria, we set the cost of vanilla inference to 1 and measure relative changes. As demonstrated below, `VISTA also outperforms [1] and [2] in efficiency`.
>
> >|Method|Relative Inference Cost↓|
> |:-|:-|
> |Vanilla|1.0|
> |ED [1]|3.13|
> |FastED [1]|1.33|
> |CT$^2$S (40%) [2]|1.43|
> |CT$^2$S (10%) [2]|1.35|
> |**Ours**|**1.27**|
>
> * **Discussion**
>
> >`The rationale and implementation of logits ensembling differ significantly between VISTA and [1][2]`. Method [1] aims to reduce visual distractions through image cropping, ensembling logits from multiple subcrops. Method [2] addresses global noise by pruning visual tokens based on the attention matrix, creating auxiliary logits. In contrast, VISTA's self-logits ensembling is motivated by the observation of early excitation behavior and is conducted across layers preceding the final layer to encourage decoding semantically meaningful tokens. Moreover, the proposed self-logits augmentation demonstrates synergy with the visual steering vector (VSV), as detailed in the *Synergy & Robustness* subsection (p.7, line 381).
>
> **Q3. What if $V_n$ is directly subtracted?**
>
> >Following your insightful suggestion, we set the visual steering vector to $V_s=-V_n$ instead of $V_s = V_p-V_n$ and validated it across various intervention strengths ($\lambda$) on LLAVA-1.5. Results below indicate that:
>
> >* Solely removing information from the negative example offers `limited benefits`.
> >* Negative-only steering is highly `sensitive to intervention strength`.
> >* Best performance of negative-only steering significantly `lags behind the original design`.
>
> >Specifically, negative-only steering collapses (F1 < 70) when $\lambda > 0.1$. These findings highlight the critical role of preserving positive information during the intervention process.
>
> >|$\lambda$|$V_s=-V_n$|$V_s=V_p-V_n$|
> |:-|:-|:-|
> ||$C_S$↓/$C_I$↓/$F1$↑|$C_S$↓/$C_I$↓/$F1$↑|
> |0.05|56.8 / 15.2 / 75.5|**50.8** / **13.6** / **76.9**|
> |0.08|52.5 / 14.9 / 75.6|**50.4** / **14.6** / **76.5**|
> |0.1|**41.6** / 14.4 / 74.2|47.4 / **13.1** / **77.8**|
> |0.11|21.4 / 12.7 / 58.1 (collapse)|**45.6** / **12.9** / **77.4**|
> |0.12|2.6 / 23.6 / 13.7 (collapse)|**41.6** / **12.2** / **77.7**|
> |0.15|0.20 / 50 / 0.1 (collapse)|**32.4** / **9.8** / **76.9**|
> |0.17|- / - / - (collapse)|**20.4** / **6.9** / **72.8**|

---

> > ### Comment · Reviewer_5sMs · 2025-04-06
> >
> > Thank you for addressing my concerns well. I agree with the results and will keep my initial score (Weak accept).

---

> > > ### Author Response · Authors · 2025-04-06
> > >
> > > Thank you for your reply! We really appreciate your feedback. Please don't hesitate to follow up questions if there is any discussion you'd like to further extend.

---

### Official Review · Reviewer_4sSv · 2025-03-24

**Overall Recommendation:** 3

**Summary:**

This paper discusses the topic of reducing hallucinations in LVLMs. The authors analyzed  LVLM’s generation dynamics through the lens of token logits ranking and proposed three types of inference issues. Then, the authors proposed a Visual Steering Vector (VSV) and a Self-Logits Augmentation (SLA) method to inhibit the hallucinations. Experiments are conducted on multiple benchmarks and demonstrated superior results over previous methods.

**Claims And Evidence:**

Yes, they are.

**Essential References Not Discussed:**

The references are satisfactory.

**Experimental Designs Or Analyses:**

Yes, experimental designs are soundness.

**Methods And Evaluation Criteria:**

Yes, appropriate evaluation criteria is applied.

**Other Comments Or Suggestions:**

Kindly review the comments mentioned earlier.

**Other Strengths And Weaknesses:**

**Strengths**
- The approach is simple yet effective.  It is a training-free approach that can be applied at inference time to existing models.
- The authors test their approach across multiple architectures, decoding strategies, and benchmarks, demonstrating a comprehensive evaluation.
- The approach demonstrates improved performance over previous methods in hallucination reduction.

**Weaknesses**
- The novelty is relatively limited. The concepts had already emerged in previous studies.
- The proposed approach increases the computation burden.
- The proposed approach incorporates some hyperparameters that may be sensitive to different benchmarks.

**Questions For Authors:**

N/A.

**Relation To Broader Scientific Literature:**

This work is particularly related to hallucination mitigation in LLM, VLM, MLLM, etc.

**Theoretical Claims:**

This work focuses more on experimental analysis rather than rigorous theoretical proofs.

---

> ### Author Rebuttal · Authors · 2025-03-26
>
> *We sincerely appreciate your thoughtful review and insightful feedback. We are glad that you find our approach __"simple yet effective"__, our evaluation is __"comprehensive"__, and demonstrate __"improved performance"__. Below we address your questions in detail*.
>
> **Q1. Regarding novelty**
>
> >Our VISTA is `novel` in terms of both its `analysis` and `methodology`. To our best knowledge:
>
> >* VISTA's token ranking analysis is `the first work` inspecting `internal token-level dynamics` of LVLMs.
> >* VISTA systematically uncovers `three novel observations about how LVLMs store and process visual information`: (1) gradual visual information loss, (2) early excitation, and (3) hidden genuine information.
> >* VISTA resembles `the first activation-space intervention method for reducing LVLM's hallucination`, effectively addressing observed phenomena.
>
> >As supported by you, VISTA is a `flexible design` allowing integration with various architectures and decoding strategies. This versatility enables VISTA to `complement ongoing research` focused on improving vision encoders, strengthening visual-textual alignment, and developing specialized decoding methods to reduce hallucinations.
>
> **Q2. Regarding computational efficiency**
>
> >VISTA functions as an efficient de-hallucination strategy that `introduces only minimal computational overhead` while demonstrating `superior efficiency compared to other baseline approaches.` We clarify this in detail below:
>
> >* **For captioning, summarization, and open-ended generation tasks**: The textual prompt remains constant (e.g., \"please describe the image...\"). VISTA forwards the textual prompt `only once` without the image, caching activations for future use (minimal overhead). For the positive example (visual + textual tokens), VISTA performs vanilla inference by forwarding tokens visual + textual once. The only additional step is constructing the visual steering vector (VSV) using the residual stream from the last input token and cached negative activations via `simple vector arithmetics` (this cost is marginal). Generation then proceeds normally under our proposed intervention. VISTA `avoids forwarding the token sequence twice` by leveraging the KV cache from input prompt tokens. Remaining computations involve logit ensembling and a few element-wise additions, resulting in mild overhead.
>
> >* **For QA-like tasks**: Textual prompts may vary per query, potentially requiring one additional forward pass for the negative instance (i.e., textual only query). However, the `textual tokens are minimal compared to image tokens`, typically constituting less than 10% of total input token sequence. Consequently, the extra computational burden remains mild compared to vanilla inference.
>
> >* **Empirically**, VISTA demonstrates `better efficiency` compared to other comparable methods (see Table 5). We kindly refer the reviewer to the panel of Reviewer 5sMs (Q2) for an extended efficiency comparison with other methods.
>
> **Q3. Regarding hyperparameter sensitivity**
>
> >VISTA includes two hyperparameters: intervention strength $\lambda$ and mixing ratio $\gamma$.
>
> > * For $\gamma$, we found $\gamma\approx0.3$ `consistently effective across all four architectures`.
>
> > * For $\lambda$, optimal values can vary by model; however, a broad range of $\lambda$ values perform effectively for each model, `as analyzed in the Synergy & Robustness subsection (p.7, line 381)` and shown in `Figs. 6, 12–14`, which demonstrate VISTA's consistent effectiveness across different $\lambda$ configurations.

---

> > ### Comment · Reviewer_4sSv · 2025-04-08
> >
> > The reviewer thanks the authors for the detailed feedback. It addressed my concerns well. I'll increase my initial score.

---

> > > ### Author Response · Authors · 2025-04-08
> > >
> > > The authors sincerely appreciate the reviewer’s thorough review and thoughtful consideration.

---

### Decision · Program_Chairs · 2025-05-01

**Decision:**

Accept (poster)

**Comment:**

This paper investigates hallucination in Large Vision-Language Models by analyzing token logit dynamics during generation and proposes a training-free, inference-time framework to mitigate these issues. After rebuttal, all reviewers voted for acceptance, and the AC acknowledges the paper's valuable contributions in both analysis and mitigation of hallucination. Concerns regarding novelty, hyperparameter tuning, and computational cost were raised, and most of them have been addressed during the rebuttal period. The Area Chair agrees that the novelty of the proposed method is somewhat limited, as the concept of steering vectors has been explored in prior works on mitigating hallucination in both LLMs and MLLMs. For example, the AC noted that the proposed Visual Steering Vector method is similar to that in [1]; Therefore, the AC suggests weak accept and please include a discussion and experimental comparison with [1] in the final version, along with addressing the other points raised by reviewers.

[1] Reducing Hallucinations in Large Vision-Language Models via Latent Space Steering, ICLR 2025.